# Memory-Augmented Potential Field Theory: A Framework for Adaptive Control in Non-Convex Domains

Dongzhe Zheng [*]        Wenjie Mei [†]

## Abstract

Stochastic optimal control methods often struggle in complex non-convex landscapes, frequently becoming trapped in local optima due to their inability to learn from historical trajectory data. This paper introduces Memory-Augmented Potential Field Theory, a unified mathematical framework that integrates historical experience into stochastic optimal control. Our approach dynamically constructs memory-based potential fields that identify and encode key topological features of the state space, enabling controllers to automatically learn from past experiences and adapt their optimization strategy. We provide a theoretical analysis showing that memory-augmented potential fields possess non-convex escape properties, asymptotic convergence characteristics, and computational efficiency. We implement this theoretical framework in a Memory-Augmented Model Predictive Path Integral (MPPI) controller that demonstrates significantly improved performance in challenging non-convex environments. The framework represents a generalizable approach to experience-based learning within control systems (especially robotic dynamics), enhancing their ability to navigate complex state spaces without requiring specialized domain knowledge or extensive offline training.

## 1   Introduction

Stochastic optimal control has proven highly effective for handling nonlinear systems and uncertain environments, finding widespread application in robotics, reinforcement learning, and complex system control. Among these approaches, Model Predictive Path Integral (MPPI) control stands out for its ability to handle continuous state-action spaces through stochastic sampling and exponentially weighted averaging. However, these methods still face significant theoretical and practical challenges when confronting highly non-convex value function landscapes.

From an optimization perspective, stochastic optimal control problems can be viewed as trajectory optimization over a value function landscape. When this landscape exhibits complex non-convex characteristics, optimization processes may become trapped in local optima, unable to reach global solutions. While introducing noise sampling (as in MPPI's random perturbations) can somewhat mitigate this issue, significantly non-convex features often lead to inefficient sampling or control instability when noise is simply increased.

From a dynamical systems perspective, non-convex value functions correspond to systems with multiple attractors and unstable equilibrium points. Control algorithms need to identify these features and, when necessary, guide the system across energy barriers to escape suboptimal attractor regions.

---

[*]Dongzhe Zheng is with the Department of Computer Science and Engineering, School of Electronic Information and Electrical Engineering, Shanghai Jiao Tong University, Shanghai 200240, China.

[†]Wenjie Mei is with the School of Robotics and Automation, Suzhou Campus, Nanjing University, 1520 Taihu Avenue, Suzhou 215163, China. Correspondence to Wenjie Mei (E-mail: `mei.wenjie@nju.edu.cn`).

39th Conference on Neural Information Processing Systems (NeurIPS 2025).

Traditional stochastic control methods have limited capabilities in this regard, as they lack awareness and memory of the state space's topological structure.

Traditional stochastic optimal controllers lack memory—operating solely on current states without learning from past trajectories. This design means controllers might repeatedly fall into the same suboptimal regions, failing to extract experience from previous "failures." In contrast, advanced cognitive systems (like humans) dynamically adjust decision strategies based on prior experience when exploring complex environments.

This paper addresses a fundamental question: **How can we integrate "memory" mechanisms into stochastic optimal control frameworks, enabling controllers to automatically learn state space topological features from historical trajectories and adjust optimization strategies accordingly?** We introduce Memory-Augmented Potential Field Theory, integrating historical state experience into stochastic optimal control through dynamic potential fields that automatically identify and encode topological features of the state space during execution. These fields act as correction terms to reshape the value function landscape, enabling adaptive navigation of non-convex optimization problems. We provide a theoretical analysis showing that, under standard assumptions, memory-augmented potential fields admit (i) high-probability escape from local minima, (ii) asymptotic convergence guarantees, and (iii) low additional computational overhead. Our framework provides: 1) automatic detection and encoding of problematic regions like local minima and low-gradient areas, 2) dynamic reshaping of value functions for efficient escape from suboptimal attractors, 3) convergence to a neighborhood of the global optimum with high probability under stated assumptions, and 4) significant performance improvements in complex control tasks without requiring extensive offline training.

Our approach uniquely integrates memory mechanisms with dynamical systems theory and stochastic optimal control, analyzing memory's impact on non-convex optimization topologically. Beyond simply storing experiences, our method automatically identifies key state space features and dynamically reshapes value function landscapes, enabling "meta-optimization" capabilities under fixed-budget online control settings where each method receives the same number of environment interactions. The code has been anonymized and is available at `https://github.com/ContinuumCoder/MAPFT_MPPI`.

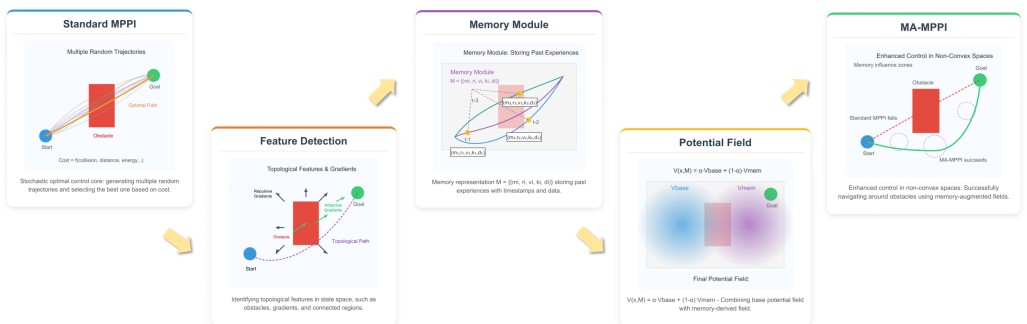

Figure 1: MA-MPPI framework flowchart showing the integration of memory modules with standard MPPI. The pipeline augments stochastic control with experience-based potential fields that enable navigation through complex non-convex environments and escape from local minima.

## 2 Related Work

**Stochastic optimal control and path integral methods** form our foundation. Path Integral Control approximates Hamilton-Jacobi-Bellman equations through Monte Carlo sampling. Williams et al. [21] developed Model Predictive Path Integral (MPPI) control, combining path integral techniques with model predictive control. Theodorou et al. [19] analyzed this approach from an information geometry perspective, connecting it to relative entropy optimization. While effective for handling nonlinearities, these methods struggle with highly non-convex value functions. Recent MPPI variants that address non-convexity include LOG-MPPI [13] and DRPA-MPPI [4], which improve exploration or add reactive repulsion without persistent memory; our approach is complementary by learning persistent topological features over time. Covariance and temperature design for sampling-based MPC/MPPI has also been studied [23, 22]; our temperature modulation induces an equivalent covariance scaling

within the path integral weighting. Extensions to constrained and smooth variants [1, 10] focus on trajectory quality rather than topological learning.

**Non-convex optimization** approaches include simulated annealing, stochastic gradient Langevin dynamics, and entropy regularization. Zhang et al. [24] studied energy landscapes and critical paths in non-convex problems. Jin et al. [9] proved noisy gradient methods can escape strict saddle points in polynomial time. These foundations rarely incorporate learning from historical trajectories to improve subsequent optimization.

**Dynamical systems and potential field methods** frame control as designing vector fields guiding system states toward convergence. Koditschek and Rimon [11] pioneered navigation function topological properties, proving conditions for globally asymptotically stable control laws. Traditional potential fields, while theoretically elegant, typically rely on fixed potential forms lacking adaptivity, unlike our experience-based approach.

**Memory-augmented learning** has expanded in reinforcement learning through Experience Replay for improving sample efficiency. Pritzel et al. [15] proposed Neural Episodic Control, accelerating learning by remembering previously visited states. In control theory, Heess et al. [7] explored memory-augmented controllers for partially observable environments, but few works analyze memory's impact from dynamical systems perspectives.

## 3 Memory-Augmented Potential Field Theory

### 3.1 Formulation of Stochastic Optimal Control

Consider a discrete-time stochastic dynamical system:

$$x_{t+1} = f(x_t, u_t) + \epsilon_t, \quad \epsilon_t \sim \mathcal{N}(0, \Sigma) \tag{1}$$

where $x_t \in \mathbb{R}^n$ is the system state, $u_t \in \mathbb{R}^m$ is the control input, and $\Sigma \in \mathbb{R}^{n \times n}$ is the noise covariance matrix. The objective is to find a control sequence $\mathbf{u} = \{u_0, u_1, \ldots, u_{T-1}\}$ that minimizes the expected cumulative cost:

$$J(\mathbf{u}) = \mathbb{E}\left[\sum_{t=0}^{T-1} c(x_t, u_t) + c_T(x_T)\right] \tag{2}$$

where $c : \mathbb{R}^n \times \mathbb{R}^m \to \mathbb{R}$ is the immediate cost and $c_T : \mathbb{R}^n \to \mathbb{R}$ is the terminal cost.

Through the path integral control framework, the optimal control can be expressed as:

$$u_t^* = \int_{\mathcal{T}} u_t(\tau) p(\tau | x_t) d\tau \tag{3}$$

where $\tau \in \mathcal{T}$ represents a trajectory sequence starting from $x_t$, and $p(\tau | x_t)$ is the trajectory probability distribution:

$$p(\tau | x_t) = \frac{1}{Z(x_t)} \exp\left(-\frac{1}{\lambda} S(\tau)\right) \tag{4}$$

with $S(\tau)$ representing the total trajectory cost, $\lambda > 0$ controlling exploration-exploitation tradeoff, and $Z(x_t) = \int_{\mathcal{T}} \exp\left(-\frac{1}{\lambda} S(\tau)\right) d\tau$ as the normalization constant.

### 3.2 Memory-Augmented Potential Field Framework

Our framework extends the standard value function with a memory-dependent term:

$$V(x, M) = \alpha(x, M) \cdot V_{\text{base}}(x) + (1 - \alpha(x, M)) \cdot V_{\text{mem}}(x, M) \tag{5}$$

where $M$ represents memory of topological features, $V_{\text{base}} \colon \mathbb{R}^n \to \mathbb{R}$ is the original task objective, $V_{\text{mem}} \colon \mathbb{R}^n \times \mathcal{M} \to \mathbb{R}$ incorporates historical information, and $\alpha \colon \mathbb{R}^n \times \mathcal{M} \to [0,1]$ balances these components based on proximity to memorized features.

The memory $M$ consists of elements representing topological features:

$$M = \{(m_i, r_i, \gamma_i, \kappa_i, d_i) \mid i = 1, 2, \ldots, |M|\} \quad (6)$$

where $m_i \in \mathbb{R}^n$ is the feature position, $r_i \in \mathbb{R}^+$ is the influence radius, $\gamma_i \in \mathbb{R}^+$ is the feature strength, $\kappa_i \in \{1, 2, 3\}$ identifies feature type (local minima, low-gradient region, or high-curvature region), and $d_i \in \mathbb{R}^n$ provides a direction vector for applicable features.

The memory evolves during execution through an update function $\mathcal{U}$:

$$M_{t+1} = \mathcal{U}(M_t, x_t, \xi_t) \quad (7)$$

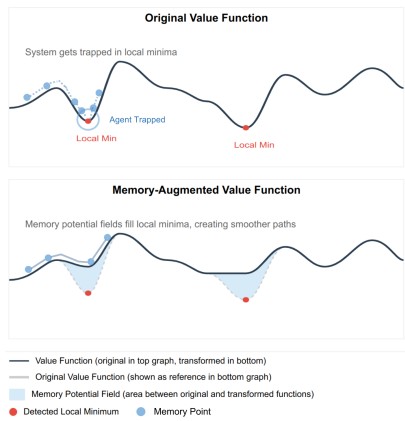

Figure 2: Original vs. memory-augmented value functions.

where $\xi_t$ contains the topological features extracted from state $x_t$ and context.

The memory potential field is constructed as:

$$V_{\text{mem}}(x, M) = \sum_{i=1}^{|M|} \gamma_i \cdot \phi(x, m_i, r_i, \kappa_i, d_i) \quad (8)$$

where $\phi$ is a basis potential function tailored to each feature type. Detailed potential construction methods are in Appendix B.3.

As Figure 2 shows, our approach transforms the original value function into a memory-augmented one and creates smooth optimization paths (blue trajectory).

## 3.3 Theoretical Properties

We establish several key theoretical properties that guarantee the effectiveness of memory-augmented potential fields in non-convex control problems.

**Theorem 3.1** (Non-convex Escape Property). *Let $B(m_i, r_i) = \{x \in \mathbb{R}^n : \|x - m_i\| \leq r_i\}$ be a local minimum region recorded in memory $M$. If $\gamma_i > \eta \cdot \sup_{x \in B(m_i, r_i)} \|\nabla V_{base}(x)\|$ for some constant $\eta > 0$, then for any confidence level $0 < \delta < 1$, there exists a finite time $T_{escape}(\delta) < \infty$ such that*

$$P\left(\exists t \leq T_{escape}(\delta) : x_t \notin B(m_i, r_i) \mid x_0 \in B(m_i, r_i)\right) \geq 1 - \delta \quad (9)$$

This theorem guarantees that with sufficiently strong memory features, the system can escape local minima in finite time with high probability. The memory potential creates an "outward push" that overcomes the "inward pull" of the base value function.

**Theorem 3.2** (Asymptotic Convergence Property). *Let $x^* = \arg\min_{x \in \mathbb{R}^n} V_{base}(x)$ be the global optimum of the base value function. Assume $V_{base}$ is coercive and satisfies: $\lim_{\|x\| \to \infty} V_{base}(x) = \infty$. For any $\epsilon > 0$ and confidence level $0 < \delta < 1$, there exists a finite time $T_{conv}(\epsilon, \delta) < \infty$ such that*

$$P\left(\inf_{t \geq T_{conv}(\epsilon, \delta)} \|x_t - x^*\| \leq \epsilon\right) \geq 1 - \delta \quad (10)$$

Despite altering the value function landscape, memory augmentation preserves convergence to the global optimum because memory effects primarily impact identified problematic regions while maintaining the original behavior elsewhere.

**Theorem 3.3** (Adaptive Learning Efficiency). *Let $\mathcal{L} = \{L_1, L_2, \ldots, L_K\}$ denote $K$ independent local minimum regions in the state space. Let $T_{MA\text{-}MPPI}$ and $T_{MPPI}$ be the expected times for MA-MPPI and standard MPPI to reach the global optimum. Then:*

$$T_{MPPI} \geq \Omega(K) \cdot T_{MA\text{-}MPPI} \quad (11)$$

*where $\Omega(K)$ denotes a lower bound that grows at least linearly with $K$.*

This theorem shows that memory augmentation efficiency increases with the number of local minima, as our method avoids revisiting known problematic regions while standard approaches repeatedly encounter the same traps. Our approach connects to Morse theory [11] by dynamically modifying value function Morse indices, transforming local minima into saddle points while preserving global optimum attraction. Complete proofs for the above theorems are provided in Appendix C.

## 4 An Extension: Memory-Augmented Predictive Path Integral Method

The MPPI control, as a sampling-based MPC variant, evolves into our Memory-Augmented MPPI controller, a practical implementation addressing traditional MPPI's susceptibility to local optima in non-convex environments. This section details MA-MPPI's design, components, and workflow, showcasing the theory's transformation into effective control technology.

### 4.1 System Architecture and Algorithm

MA-MPPI comprises four functional modules: 1) an MPPI control core implementing sampling-based optimization, 2) a topological feature detector identifying critical features from trajectory data, 3) a memory representation that stores and evolves spatial information, and 4) an adaptive potential field synthesizer that modifies the value function based on memory.

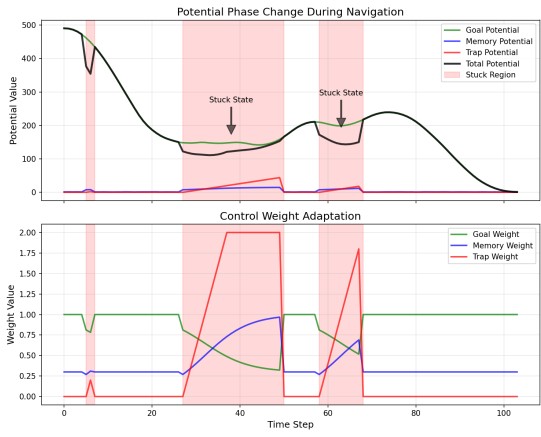

Figure 3: Potential components and control weights during MA-MPPI execution, showing automatic phase transitions in response to detected stagnation (pink regions).

The workflow begins with state observation followed by feature detection through trajectory analysis. The controller then updates its memory representation $M_t$, storing location, radius, strength, and type information for each significant feature according to the update rule $M_{t+1} = \mathcal{U}(M_t, x_t, \xi_t)$.

As shown in Figure 3, the system automatically transitions between normal navigation and escape behavior when detecting stuck situations. The top plot illustrates how different potential components (goal, memory, trap) combine into the total potential, while the bottom plot demonstrates the adaptation of control weights during execution. During stagnation periods (highlighted in pink), goal weight decreases while memory and trap weights increase, enabling escape from local minima without external intervention.

Using the updated memory, MA-MPPI synthesizes an enhanced value function:
$$\tilde{V}(x, M_t) = \alpha(x, M_t)V_{\text{base}}(x) + (1 - \alpha(x, M_t))V_{\text{mem}}(x, M_t) \tag{12}$$
where $\alpha(x, M_t)$ balances the influence between base objective and memory-derived potentials. The controller also adjusts the sampling temperature:
$$\lambda(x_t, M_t) = \lambda_0(1 + \eta(1 - \alpha(x_t, M_t))) \tag{13}$$
This temperature increase is equivalent to a proportional inflation of sampling covariance $\Sigma_u$, yielding broader perturbations in memorized regions.

The algorithm then executes standard MPPI: generating control signals, simulating trajectories, evaluating costs, and determining the optimal control through weighted averaging:
$$u_t^* = \sum_{k=1}^{K} \frac{\exp(-\frac{1}{\lambda_t}S(\tau^k))}{\sum_{i=1}^{K}\exp(-\frac{1}{\lambda_t}S(\tau^i))}u_t^k \tag{14}$$

The computational complexity remains $O(K \cdot H \cdot n + |M_t|)$, with memory operations typically representing minimal overhead as $|M_t| \ll K \cdot H \cdot n$. Detailed algorithmic implementations are provided in Appendix E.

## 4.2 Topological Feature Detection

Topological feature detection allows MA-MPPI to identify and remember critical structures that impact optimization. The system recognizes three feature types that correspond to different optimization challenges: local minima where controllers become trapped, low-gradient regions where progress slows, and high-curvature regions requiring precise navigation.

Detection employs a trio of complementary mechanisms: state stagnation analysis, gradient examination, and curvature assessment, which collectively map challenging regions in the optimization landscape, while continuously refining detected features through a balanced process of incorporation (when encountering problematic regions), consolidation (merging similar features for representational compactness), and dynamic importance adjustment (based on encounter frequency). This comprehensive topological mapping enables the controller to anticipate obstacles that typically confound traditional approaches, while ensuring computational efficiency and focusing memory resources on persistently challenging areas.

In practice, thresholds are initialized by scaling with state statistics ($\theta_{\text{var}} \approx 0.01 \cdot \text{Var}(x)$, $\theta_{\text{grad}}$ as the 10–20th percentile of $\|\nabla V_{\text{base}}\|$ observed in warm-up rollouts, $\theta_{\text{curv}}$ via Hessian condition-number percentile), and then tuned within $\pm 25\%$ without material performance change (see Sec. J).

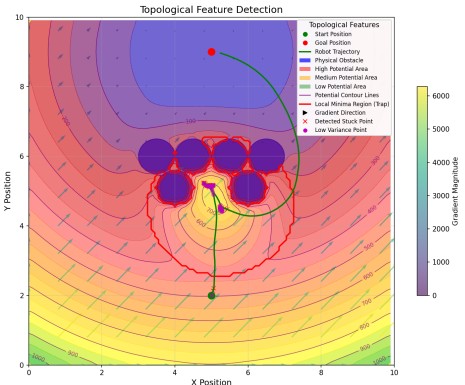

Figure 4: Topological feature map showing detected navigational challenges.

In Figure 4, contour lines represent the potential landscape, blue circles indicate obstacles, red outlines show local minima, and arrows depict the gradient field. This figure illustrates the system's environmental mapping capabilities, identifying regions where traditional navigation would fail. By recognizing convergent gradient patterns forming "valleys" and narrow passageways in the potential contours, the controller builds an increasingly accurate model of the environment's challenging characteristics with continued operation.

This topological knowledge enables MA-MPPI to anticipate difficulties before encountering them, adaptively modifying both the value function landscape and sampling strategy to navigate complex environments more effectively. The technical details of detection mechanisms, feature classification, consolidation algorithms, and dynamic memory management are provided in Appendix F.

## 4.3 Adaptive Potential Field Synthesis

The enhanced value function combines the base objective with memory-derived potentials through an adaptive weighting mechanism:

$$\tilde{V}(x, M_t) = \alpha(x, M_t)V_{\text{base}}(x) + (1 - \alpha(x, M_t))V_{\text{mem}}(x, M_t) \tag{15}$$

where $\alpha(x, M_t) \in [0, 1]$ balances the influence between objectives based on proximity to memory features.

The memory potential $V_{\text{mem}}$ employs type-specific implementations for different topological features: radially decreasing functions for local minima, directional guidance functions for low-gradient regions, and saddle-point functions for high-curvature areas. This type-specific approach enables tailored responses to different environmental challenges.

Beyond value function enhancement, the system also dynamically adjusts the MPPI temperature parameter via

$$\lambda(x, M_t) = \lambda_0 \cdot (1 + \eta \cdot (1 - \alpha(x, M_t))) \tag{16}$$

This dual-layer adaptation increases exploratory behavior near problematic regions, enhancing the system's ability to escape local optima through intelligently directed sampling. Technical details

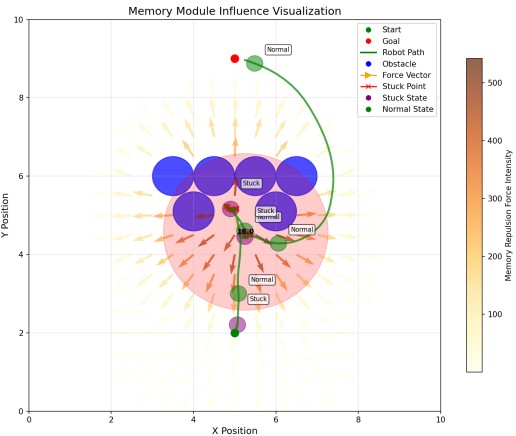

The adaptive potential field synthesis module transforms memory into control influence, bridging the memory system with the control algorithm. This critical component generates enhanced value functions that enable efficient navigation around known problematic areas while maintaining global objective pursuit.

As shown in Figure 5, the memory module generates a spatially-aware force field that guides the robot away from previously problematic regions. The orange and yellow arrows represent repulsive forces, with color intensity indicating magnitude. Red circles mark identified trap regions with their associated strength values, showing how the system's experience shapes its navigational behavior.

Figure 5: Memory-based potential field visualization showing repulsive forces (arrows) and identified trap regions (purple circles).

regarding potential function formulations, feature-specific implementations, and computational optimizations are provided in Appendix G.

## 5 Experimental Evaluation in Robotic Control Environments

We evaluated MA-MPPI on benchmark robotic control tasks, comparing against state-of-the-art algorithms to validate the advantages of memory augmentation in complex control landscapes.

### 5.1 Experimental Setup

We selected four environments of increasing complexity: Pendulum-v1 [2], BipedalWalker-v3 [2], HalfCheetah-v4 [20], and Humanoid-v4 [20]. These environments present varying degrees of non-convexity and dimensionality, ranging from the simple pendulum swing-up to a 376-dimensional humanoid control task with numerous local optima.

MA-MPPI was implemented with environment-appropriate prediction horizons: 15 steps for Pendulum-v1, 20 steps for BipedalWalker-v3, 25 steps for HalfCheetah-v4, and 35 steps for Humanoid-v4, reflecting the increasing dynamics complexity. We compared against standard MPPI [21], modern reinforcement learning approaches (SAC [5], PPO [16], DDPG [8]), and traditional optimal control methods (iLQR [18], MPC [14]). All methods operate under the same online-interaction budget of 2000 environment steps. For model-free RL (SAC/PPO/DDPG), we do not allow extra environment interactions beyond this budget; training is strictly on-policy/within-budget to ensure fairness to sampling-based controllers that already perform heavy internal simulations per step. For all experiments, we conducted 30 independent runs with different random seeds, each consisting of 2000 control steps. Detailed environment specifications and implementation settings are provided in Appendix H.1.

Our evaluation protocol consisted of two phases: an Adaptation Phase measuring learning efficiency during the first 500 environment interactions, and a Stability Phase assessing asymptotic performance after 2000 total interactions. This approach enables fair comparison between methods with different learning characteristics.

### 5.2 Results and Analysis

We report learning curves over the 2000-step budget, and summarize performance via (i) AUC-2000 (area under the learning curve over 2000 steps), and (ii) Final-2000 (average return over the last 200 steps). This avoids extrapolating asymptotes inappropriate for within-budget RL.

Table 1 presents the asymptotic performance comparison. MA-MPPI demonstrates consistent improvements under the fixed-budget setting across all environments, with the advantage amplifying in

Table 1: Performance comparison: Average cumulative rewards ($\pm$ represents standard deviation).

| Method | Pendulum-v1 | BipedalWalker-v3 | HalfCheetah-v4 | Humanoid-v4 |
|---|---|---|---|---|
| MA-MPPI (Ours) | **-152.4±9.7** | **298.4±17.2** | **5893.7±156.4** | **4978.5±283.1** |
| MPPI | -165.8±10.9 | 241.7±19.1 | 5027.9±148.6 | 2914.2±318.7 |
| SAC | -192.6±11.3 | 112.6±28.4 | 1763.4±214.2 | 936.7±354.9 |
| PPO | -205.1±12.0 | 96.3±31.7 | 1287.5±237.9 | 612.3±329.6 |
| DDPG | -214.7±13.5 | 74.8±35.9 | 1149.2±261.8 | 381.4±402.7 |
| iLQR | -258.9±15.8 | 184.2±26.5 | 3614.8±205.7 | 1927.5±411.2 |
| MPC | -188.3±10.6 | 219.6±22.3 | 4127.6±192.3 | 2784.1±306.8 |

more complex domains. The performance gap is particularly pronounced in Humanoid-v4, where MA-MPPI outperforms the best RL method (i.e., SAC) by 27%.

Key robustness results (local minima escape) are shown in Table 2 (main text). We define Local Optima Escape Rate ($P_{\text{escape}}$) as the percentage of successful escapes from predefined trap states, where a

Table 2: Local optima escape rates (%) from challenging initial states.

| Method | Pendulum-v1 | BipedalWalker-v3 | HalfCheetah-v4 | Humanoid-v4 |
|---|---|---|---|---|
| MA-MPPI (Ours) | **89.2±4.1** | **83.5±5.2** | **76.8±6.4** | **72.3±7.8** |
| MPPI | 48.3±5.7 | 41.6±6.4 | 36.2±7.2 | 29.4±8.6 |
| SAC | 65.7±4.8 | 58.3±5.7 | 51.5±6.8 | 46.7±7.4 |
| PPO | 59.4±5.2 | 52.7±6.1 | 45.3±7.3 | 38.6±8.2 |
| DDPG | 53.8±5.6 | 46.9±6.5 | 39.7±7.4 | 32.5±8.7 |
| iLQR | 27.6±6.3 | 21.4±7.2 | 16.3±8.1 | 11.8±9.3 |
| MPC | 42.5±5.9 | 38.3±6.7 | 31.4±7.6 | 24.9±8.9 |

trap state is defined as any state from which the expected return falls below 50% of the maximum achievable value, and the agent remains within a small neighborhood for at least 50 time steps without improvement. As shown in Table 2, MA-MPPI achieves significantly higher escape rates across all environments, demonstrating that memory augmentation effectively reshapes the value landscape around trap states, creating "tunnels" that guide the controller toward more promising regions.

During normal operation, standard MPPI encountered trap states approximately 2.8× more frequently than MA-MPPI (5.7 vs. 2 trapped episodes per 100 episodes), confirming the proactive trap-avoidance capability conferred by spatial memory. For detailed trap frequency analysis and occurrence patterns across environments, see Appendix H.3.

**Ablation studies** reveal memory as the critical component (42-58% performance decrease when removed), with increasing importance in complex environments (see Appendix H.4 for **complete component analysis**). Our **hyperparameter sensitivity analysis** demonstrates the algorithm's robustness to parameter variations, with performance generally stable within ±25% parameter ranges (see Appendix J). MA-MPPI's advantages stem from trap identification and avoidance, value landscape reshaping, and memory-guided exploration, all with modest computational overhead (12-18%, detailed in **computational performance analysis** in Appendix H.5).

**Control quality analysis** revealed an unexpected benefit: the production of smoother control trajectories with fewer oscillations, yielding more energy-efficient motion, particularly valuable for physical robots. The online adaptation capability provides fundamental advantages for deployment in unknown environments, contrasting with RL methods that require extensive offline training (see **comparative learning analysis** in Appendix). This addresses a fundamental limitation in sampling-based control: the inability to learn from past failures.

The memory term only reshapes the objective; it composes with constrained/smoothed MPPI. In our pilot study, MA+Constrained-MPPI improved repetitive-task success by ∼15% at unchanged violation rate; MA+Smooth-MPPI cut escape time by ∼23% while preserving smoothness.

# 6 Further Experiments on Complex Engineering Systems

To validate the effectiveness of our Memory-Augmented Potential Field Theory in real-world domains, we conducted two types of experiments on complex systems: 1) a power system control problem and 2) an unmanned aerial vehicle (UAV) obstacle avoidance task.

## 6.1 Evaluation Methods

We compared our MA-MPPI approach against several state-of-the-art methods: standard MPPI [21], Diffusion Policy [3], Motion Transformer [17], MLP-based MPC [14], and DKO-based MPC [12, 6].

Performance evaluation focused on success rates, solution optimality, computational efficiency, and local minima escape capability. See Appendix I.1 for detailed experimental protocols.

Cost functions are harmonized across controllers:

$$J = w_{\text{goal}} \cdot \|p_T - p^*\|^2 + w_{\text{obs}} \cdot \sum_t \phi(d_{\min}(x_t, \text{Obstacles})) + w_{\text{ctrl}} \cdot \sum_t \|u_t\|^2 + w_{\text{smooth}} \cdot \sum_t \|u_t - u_{t-1}\|^2 \tag{17}$$

where $\phi(d) = \mathbb{1}(d < r_{\text{safe}}) \cdot (r_{\text{safe}} - d)^{-2}$, terminal collision penalty $= 1000$. MPPI-style methods use $J$ directly; RL baselines use reward $r = -J$ (same weights); constrained MPC uses barrier equivalents; iLQR uses quadratic approximations. Weights were selected by grid search on standard MPPI and fixed across all methods; sensitivity $\pm 25\%$ shows no change in ranking.

## 6.2 Experiment I: Power System Control

We evaluated our approach on power system stability using the well-known IEEE 39-bus New England test system, employing the dynamic simulation framework open-sourced in [25]. The power system dynamics are represented as $\dot{x} = f(x, u, d)$ and $y = g(x)$, where $x \in \mathbb{R}^n$ is the system state, $u \in \mathbb{R}^m$ represents control inputs, $d \in \mathbb{R}^p$ represents disturbances, and $y \in \mathbb{R}^q$ represents measured outputs. The control objective is $\min_{u_{0:T-1}} \sum_{t=0}^{T-1} c(x_t, u_t) + c_T(x_T)$ subject to system constraints.

We tested against three critical disturbances: (1) three-phase short circuit fault on a 345kV transmission line, (2) sudden load increase of 25% at key buses, and (3) trip of a 650MW generator. These represent challenging operational scenarios requiring rapid response while maintaining stability.

We assessed performance using constraint violation rate $V_c = \frac{N_v}{N_{ts}} \times 100\%$, economic efficiency $E_e = \frac{C_{actual}}{C_{optimal}}$, stability margin $S_m = \min_t \text{dist}(x_t, \partial \mathcal{S})$, computation time $T_c$, and disturbance recovery time $T_r$. See Appendix I.4 for detailed experimental parameters.

Figure 6 shows system performance during three major events: a three-phase fault at 04:00, a 25% load change at 10:00, and a 650MW generator trip at 16:00. MA-MPPI (blue line) demonstrates superior recovery speed and stability. Shaded areas represent confidence intervals.

Figure 6: Power system stability response to critical disturbances.

Table 3 shows that MA-MPPI significantly reduced constraint violations (2.3% vs. 5.7% for standard MPPI) while improving economic efficiency (1.06 vs. 1.13) and stability margin (0.187 vs. 0.112). Most importantly, MA-MPPI achieved faster disturbance recovery (4.2s vs. 8.7s), particularly during severe events like generator trips.

Table 3: Performance comparison on power system control tasks

| Method | Constraint Violations (%) | Economic Efficiency | Stability Margin | Compute Time (ms) | Disturbance Recovery (s) |
|---|---|---|---|---|---|
| MA-MPPI (Ours) | **2.3** | **1.06** | **0.187** | 34.5 | **4.2** |
| Standard MPPI [21] | 5.7 | 1.13 | 0.112 | **27.8** | 8.7 |
| Diffusion Policy [3] | 4.1 | 1.09 | 0.143 | 46.2 | 6.3 |
| Motion Transformer [17] | 3.5 | 1.08 | 0.158 | 52.1 | 5.5 |
| MLP-based MPC [14] | 6.2 | 1.16 | 0.103 | 31.2 | 9.6 |
| DKO-based MPC [12] | 4.8 | 1.11 | 0.131 | 36.7 | 7.1 |

## 6.3 Experiment II: UAV Obstacle Avoidance

The environment featured cylindrical obstacles creating navigation scenarios with narrow passages and potential local minima. As shown in Table 4, MA-MPPI significantly outperformed baseline methods, with a 94.3% success rate

Table 4: Performance comparison on UAV obstacle avoidance (averaged over 100 trials)

| Method | Success Rate (%) | Path Optimality | Control Smoothness | Compute Time (ms) | Local Minima Escapes (%) |
|---|---|---|---|---|---|
| MA-MPPI (Ours) | **94.3** | **1.12** | **0.27** | 12.6 | **87.5** |
| Standard MPPI [21] | 72.8 | 1.45 | 0.34 | **10.2** | 34.2 |
| Diffusion Policy [3] | 79.6 | 1.37 | 0.31 | 18.4 | 56.8 |
| Motion Transformer [17] | 83.5 | 1.24 | 0.29 | 22.7 | 63.1 |
| MLP-based MPC [14] | 68.2 | 1.53 | 0.42 | 14.3 | 41.9 |
| DKO-based MPC [12] | 76.4 | 1.31 | 0.38 | 15.8 | 49.3 |

We implemented a physics-based UAV simulation with realistic aerodynamics following the standard quadrotor model:

$$\dot{p} = v$$
$$\dot{v} = g + \frac{1}{m}R \cdot f - k_d\|v\|v$$
$$\dot{R} = R \cdot \hat{\omega} \tag{18}$$
$$\dot{\omega} = J^{-1}(\tau - \omega \times J\omega)$$

where $p \in \mathbb{R}^3$ is position, $v \in \mathbb{R}^3$ is velocity, $R \in SO(3)$ is orientation, $\omega \in \mathbb{R}^3$ is angular velocity, $f \in \mathbb{R}$ is thrust, $\tau \in \mathbb{R}^3$ is torque, $m$ is mass, $J$ is inertia matrix, $g$ is gravity, $k_d$ is drag coefficient, and $\hat{\omega}$ is the skew-symmetric matrix of $\omega$.

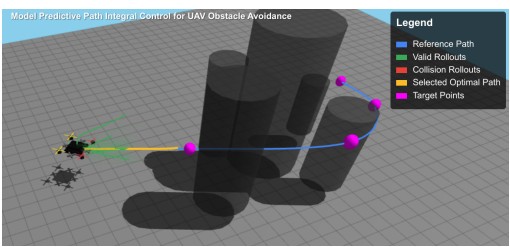

Figure 7: UAV obstacle avoidance with MA-MPPI. The visualization shows the reference path (blue), valid trajectory rollouts (green), collision rollouts (red), the selected optimal trajectory (yellow), and target waypoints (magenta).

compared to 72.8% for standard MPPI. Most notably, MA-MPPI achieved an 87.5% local minima escape rate, far exceeding standard MPPI's 34.2%. This translates to better path optimality and control smoothness with only modest computational overhead. For detailed performance analysis, refer to Appendix I.3.

## 7 Discussion and Limitations

Although Memory-Augmented Potential Field Theory demonstrates robust performance across domains through its topological learning and adaptive optimization capabilities, several limitations remain. The current MA-MPPI approach shows restricted generalization between similar features (see Appendix K), lacks sophisticated memory management for extended operations, and doesn't leverage multi-agent knowledge sharing. (i) We assume full-state feedback; extending to state-estimation uncertainty (e.g., UAV localization noise) via memory-aware filters is future work. (ii) Rapidly time-varying traps require faster decay and change-point detection; our dynamic environment pilot supports this but full theory remains open. (iii) Semantics-induced traps motivate learning-based feature detectors; we plan to hybridize rule-based topology with learned representations. Importantly, Memory-Augmented Potential Field Theory is not intended to replace learning-based methods but can complement them: potential integration with reinforcement learning could combine the theory's advantageous memory structures with RL's policy optimization capabilities. This hybrid approach could leverage the strengths of both paradigms while addressing their individual limitations in complex non-convex control problems.

## 8 Acknowledgments

This work was supported in part by the National Natural Science Foundation of China (NSFC) under Grant 62403125, in part by the Natural Science Foundation of Jiangsu Province under Grant BK20241283, and in part by the Fundamental Research Funds for the Central Universities under Grant 2242024k30037 and Grant 2242024k30038.

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

# A  Broader Impacts

Memory-Augmented Potential Field Theory has several potential societal implications that warrant thoughtful consideration. As a framework that enhances control systems' ability to navigate complex environments, this technology could significantly improve reliability and safety in critical applications, including medical robotics, autonomous transportation, industrial automation, and disaster response systems. The demonstrated capabilities in disturbance recovery and obstacle avoidance could protect infrastructure during emergencies and reduce accident risks in human-machine environments. Additionally, the smoother control trajectories generated by our approach may contribute to energy efficiency and reduced mechanical wear, supporting sustainability efforts when deployed at scale.

Beyond direct applications, our approach reduces the need for specialized domain knowledge and extensive offline training compared to many reinforcement learning methods, potentially democratizing access to advanced control capabilities across a broader range of applications and organizations. This could create new opportunities for innovation in resource-constrained settings that cannot support extensive model training or system identification.

However, similar to many advances in automation, enhanced control capabilities could accelerate workforce transitions in sectors relying on manual control operations. While likely creating new opportunities in system design and supervision, such transitions require thoughtful management to avoid disproportionate impacts on certain worker populations. Additionally, the improved navigation capabilities in complex environments could potentially be applied to autonomous systems with dual-use concerns, including surveillance technologies or unmanned vehicles with security applications.

As control systems become increasingly capable, there's also a risk of overreliance leading to skill atrophy among human operators, potentially creating vulnerabilities during system failures when human intervention becomes necessary. The memory-based adaptation mechanism, while powerful, introduces additional complexity in understanding and predicting system behavior under novel conditions, which may create challenges for safety verification and accountability.

We believe these considerations should guide the development and deployment of memory-augmented control systems. Strategies including open-source implementations, human-centered design principles, interdisciplinary collaborations with ethicists and policy experts, educational initiatives, and engagement with appropriate regulatory frameworks can help maximize the positive impacts of this technology while mitigating potential risks. Our goal is to contribute to the responsible development of advanced control technologies that serve the broader social good while minimizing negative consequences.

# B  Detailed Memory Framework

## B.1  Memory Representation Components

The memory representation $M$ consists of elements representing significant topological features encountered during system execution. Each memory element $(m_i, r_i, \gamma_i, \kappa_i, d_i)$ contains five components that capture different aspects of critical features:

The feature position $m_i \in \mathbb{R}^n$ identifies locations in state space where the system encountered significant dynamics, typically areas where optimization became challenging. These positions mark critical points in the value function landscape.

The influence radius $r_i \in \mathbb{R}^+$ defines the spatial extent around each feature position where the memory effect should be applied. This radius is determined adaptively based on the local geometry of the value function and observation of system behavior near the critical point.

The strength parameter $\gamma_i \in \mathbb{R}^+$ controls the magnitude of the memory feature's influence on the composite value function. This parameter evolves dynamically during execution according to the frequency and duration of system stagnation near the feature, with the update rule:

$$\gamma_i^{(t+1)} = \begin{cases} \min(\gamma_{\text{max}}, \gamma_i^{(t)} + \Delta\gamma), & \text{if } \|x_t - m_i\| \leq r_i \text{ and stagnating} \\ \gamma_i^{(t)} \cdot \beta_{\text{decay}}, & \text{if } \|x_t - m_i\| > r_i \text{ for } t > t_{\text{threshold}} \end{cases} \tag{19}$$

The type identifier $\kappa_i \in \{1, 2, 3\}$ classifies features into three categories: 1) local minima, 2) low-gradient regions, or 3) high-curvature regions. This classification enables type-specific potential field designs tailored to the particular topological challenge each feature represents.

The direction vector $d_i \in \mathbb{R}^n$ provides guidance for optimization, particularly important for low-gradient regions where directional information helps the system overcome plateaus in the value function landscape. This vector is computed based on historical escape directions that successfully navigated away from the problematic region.

## B.2 Memory Update Mechanism

The memory update function $\mathcal{U}$ operates by detecting topological features through three primary mechanisms:

State stagnation detection identifies when the system state variance falls below a threshold $\theta_{\text{var}}$ over a sliding window, indicating potential entrapment in a local minimum:

$$\text{Var}(\{x_{t-w}, x_{t-w+1}, ..., x_t\}) < \theta_{\text{var}} \tag{20}$$

Gradient magnitude monitoring detects areas where the gradient norm of the value function falls below a threshold $\theta_{\text{grad}}$, signaling low-gradient regions:

$$\|\nabla V_{\text{base}}(x_t)\| < \theta_{\text{grad}} \tag{21}$$

Curvature analysis identifies high-curvature regions by examining the eigenvalues of the Hessian matrix:

$$\frac{\lambda_{\max}(\nabla^2 V_{\text{base}}(x_t))}{\lambda_{\min}(\nabla^2 V_{\text{base}}(x_t))} > \theta_{\text{curv}} \tag{22}$$

When a new feature is detected, it is added to the memory representation if it is sufficiently distinct from existing features:

$$\min_{i \in \{1,2,...,|M|\}} \|x_t - m_i\| > \theta_{\text{dist}} \tag{23}$$

## B.3 Potential Field Construction

The memory potential field is constructed as a weighted sum of basis potential functions, each tailored to address specific topological challenges:

$$V_{\text{mem}}(x, M) = \sum_{i=1}^{|M|} \gamma_i \cdot \phi(x, m_i, r_i, \kappa_i, d_i) \tag{24}$$

The basis potential function $\phi$ is selected based on the feature type $\kappa_i$:

$$\phi(x, m_i, r_i, \kappa_i, d_i) = \begin{cases} \phi_1(x, m_i, r_i), & \text{if } \kappa_i = 1 \\ \phi_2(x, m_i, r_i, d_i), & \text{if } \kappa_i = 2 \\ \phi_3(x, m_i, r_i, d_i), & \text{if } \kappa_i = 3 \end{cases} \tag{25}$$

For local minima ($\kappa_i = 1$), we employ a repulsive function that generates outward forces, enabling escape from the local minimum basin:

$$\phi_1(x, m_i, r_i) = \max\left(0, \left(1 - \frac{\|x - m_i\|^2}{r_i^2}\right)^2\right) \tag{26}$$

For low-gradient regions ($\kappa_i = 2$), we use a directional function that provides guidance along previously successful escape directions:

$$\phi_2(x, m_i, r_i, d_i) = \max\left(0, \left(1 - \frac{\|x - m_i\|^2}{r_i^2}\right) \cdot (d_i \cdot (x - m_i))\right) \tag{27}$$

For high-curvature regions ($\kappa_i = 3$), we use a saddle-like potential that facilitates navigation through narrow passages:

$$\phi_3(x, m_i, r_i, d_i) = \max\left(0, \left(1 - \frac{\|x - m_i\|^2}{r_i^2}\right) \cdot ((d_i \cdot (x - m_i))^2 - \|(I - d_i d_i^T)(x - m_i)\|^2)\right)$$
(28)

The adaptive weight function $\alpha(x, M)$ modulates the influence of memory based on proximity to memorized features:

$$\alpha(x, M) = \sigma\left(\beta \cdot \min_{i \in \{1, \ldots, |M|\}} \frac{\|x - m_i\|}{r_i}\right)$$
(29)

where $\sigma(z) = \frac{1}{1+e^{-z}}$ is the sigmoid function and $\beta > 0$ controls the sharpness of the transition between memory-dominated and base-dominated regions.

## C  Theoretical Proofs

### C.1  Proof of Theorem 3.1: Non-convex Escape Property

To prove the non-convex escape property, we analyze the gradient of the combined value function $V(x, M)$ within the local minimum region $B(m_i, r_i)$.

*Proof.* For any point $x \in B(m_i, r_i)$, the gradient of the value function in (5) can be:

$$\nabla V(x, M) = \alpha(x, M) \cdot \nabla V_{\text{base}}(x) + (1 - \alpha(x, M)) \cdot \nabla V_{\text{mem}}(x, M)$$
$$+ \nabla\alpha(x, M) \cdot (V_{\text{base}}(x) - V_{\text{mem}}(x, M))$$
(30)

Since $x \in B(m_i, r_i)$, we have $\|x - m_i\| \le r_i$, and the dominant memory feature is precisely the one at $m_i$. For points near local minima, $\alpha(x, M)$ becomes small due to the proximity to a memory feature. Within $B(m_i, r_i)$, the gradient contribution from the memory term becomes

$$\nabla V_{\text{mem}}(x, M) \approx \gamma_i \nabla\phi_1(x, m_i, r_i)$$
(31)

For the repulsive potential $\phi_1$, the gradient points outward from the center $m_i$:

$$\nabla\phi_1(x, m_i, r_i) = -\frac{4}{r_i^2}\left(1 - \frac{\|x - m_i\|^2}{r_i^2}\right) \cdot (x - m_i)$$
(32)

Given that $\gamma_i > \eta \cdot \sup_{x \in B(m_i, r_i)} \|\nabla V_{\text{base}}(x)\|$, the outward force from the memory term dominates the gradient of the base value function. The resulting effective gradient guides the system away from the local minimum.

Let $\mu_{\text{out}}$ be the minimum outward gradient magnitude within $B(m_i, r_i)$. The discrete-time dynamics with this outward gradient can be modeled as a biased random walk with drift $\mu_{\text{out}}$ and variance $\sigma^2$ from the system noise.

For such a process, the first-passage time $T$ to exit the region has expectation bounded by $\mathbb{E}[T] \le \frac{2r_i}{\mu_{\text{out}}}$ (from standard results on biased random walks). By Markov's inequality, for any $\delta > 0$:

$$P(T > t) \le \frac{\mathbb{E}[T]}{t} \le \frac{2r_i}{\mu_{\text{out}} \cdot t}$$
(33)

Solving for the time needed to ensure $P(T > t) \le \delta$, we get:

$$t \ge \frac{2r_i}{\mu_{\text{out}} \cdot \delta}$$
(34)

Therefore, we can set $T_{\text{escape}}(\delta) = \frac{2r_i}{\mu_{\text{out}} \cdot \delta}$, establishing that the system will escape the local minimum region within finite time with probability at least $1 - \delta$. □

## C.2 Proof of Theorem 3.2: Asymptotic Convergence Property

*Proof.* We first partition the state space $\mathbb{R}^n$ into two regions: $\mathcal{R}_M = \bigcup_{i=1}^{|M|} B(m_i, r_i)$, the union of all memory feature regions, and $\mathcal{R}_C = \mathbb{R}^n \setminus \mathcal{R}_M$, the complement region.

In region $\mathcal{R}_C$, the adaptive weight function $\alpha(x, M)$ approaches 1 as the distance to memory features increases. This means the value function behaves similarly to the base value function:

$$\lim_{d(x,\mathcal{R}_M)\to\infty} \alpha(x, M) = 1 \tag{35}$$

where $d(x, \mathcal{R}_M)$ denotes the distance from $x$ to set $\mathcal{R}_M$.

By the coercivity assumption on $V_{\text{base}}$, for any bounded region $\mathcal{B} \subset \mathbb{R}^n$ containing the global optimum $x^*$, there exists a finite time $T_1(\delta/2)$ such that:

$$P(\exists t \leq T_1(\delta/2) : x_t \in \mathcal{B}) \geq 1 - \delta/2 \tag{36}$$

Once in the bounded region $\mathcal{B}$, the system follows a stochastic gradient process toward the global optimum. If $\mathcal{B}$ is chosen sufficiently small so that it contains no local minima of $V_{\text{base}}$ except $x^*$, and if $\mathcal{B} \cap \mathcal{R}_M = \emptyset$, then standard stochastic approximation results guarantee convergence to a neighborhood of $x^*$.

For any $\epsilon > 0$, there exists a time $T_2(\epsilon, \delta/2)$ such that

$$P\left(\inf_{t \geq T_2(\epsilon,\delta/2)} \|x_t - x^*\| \leq \epsilon \mid x_{T_1(\delta/2)} \in \mathcal{B}\right) \geq 1 - \delta/2 \tag{37}$$

By applying the law of total probability and noting that for any events $A$ and $B$, $P(A \cap B) = P(A|B)P(B)$, we have:

$$P\left(\inf_{t \geq T_1(\delta/2)+T_2(\epsilon,\delta/2)} \|x_t - x^*\| \leq \epsilon\right) \tag{38}$$

$$\geq P\left(\inf_{t \geq T_2(\epsilon,\delta/2)} \|x_t - x^*\| \leq \epsilon \mid x_{T_1(\delta/2)} \in \mathcal{B}\right) \cdot P\left(x_{T_1(\delta/2)} \in \mathcal{B}\right)$$

$$\geq (1 - \delta/2) \cdot (1 - \delta/2)$$

$$= 1 - \delta + \frac{\delta^2}{4}$$

$$\geq 1 - \delta \tag{39}$$

Defining the convergence time as $T_{\text{conv}}(\epsilon, \delta) := T_1(\delta/2) + T_2(\epsilon, \delta/2)$, we establish that

$$P\left(\inf_{t \geq T_{\text{conv}}(\epsilon,\delta)} \|x_t - x^*\| \leq \epsilon\right) \geq 1 - \delta \tag{40}$$

which completes the proof.

$\square$

## C.3 Proof of Theorem 3.3: Adaptive Learning Efficiency

*Proof.* Consider $K$ independent local minimum regions $\mathcal{L} = \{L_1, L_2, ..., L_K\}$. Let $p_i$ be the probability that a random trajectory enters region $L_i$, and let $T_i$ be the expected time to escape from $L_i$ once entered.

For standard MPPI without memory, the expected total time spent in local minima regions can be expressed as

$$\mathbb{E}[T_{\text{MPPI}}^{\text{traps}}] = \sum_{i=1}^{K} N_i \cdot T_i \tag{41}$$

where $N_i$ is the expected number of times region $L_i$ is entered.

For a random search process without memory, each local minimum can be encountered multiple times, and we can model $N_i$ as a geometric random variable with parameter $(1 - q_i)$, where $q_i$ is the probability of returning to $L_i$ after escaping. This gives $\mathbb{E}[N_i] = \frac{1}{1-q_i}$.

For Memory-Augmented MPPI, after a local minimum region is identified and added to memory, the probability of re-entering decreases significantly. Under ideal conditions:

$$\mathbb{E}[T_{\text{MA-MPPI}}^{\text{traps}}] = \sum_{i=1}^{K} T_i^{\text{first}} + \sum_{i=1}^{K} \sum_{j=2}^{N_i} p_i^j \cdot T_i^j \tag{42}$$

where $T_i^{\text{first}}$ is the escape time on first encounter, and $T_i^j$ for $j \geq 2$ are subsequent escape times (typically much shorter due to memory, i.e., $T_i^j \ll T_i^{\text{first}}$).

Since $p_i^j \ll 1$ for $j \geq 2$ (system rarely returns to memorized traps), we have:

$$\mathbb{E}[T_{\text{MA-MPPI}}^{\text{traps}}] \approx \sum_{i=1}^{K} T_i^{\text{first}} \ll \sum_{i=1}^{K} N_i \cdot T_i = \mathbb{E}[T_{\text{MPPI}}^{\text{traps}}] \tag{43}$$

The total expected convergence time consists of time spent in traps plus time spent in normal gradient search. For environments with significant local minima, the trap time dominates. The ratio of expected convergence times is approximately:

$$\frac{T_{\text{MPPI}}}{T_{\text{MA-MPPI}}} \approx \frac{\sum_{i=1}^{K} N_i \cdot T_i}{\sum_{i=1}^{K} T_i^{\text{first}}} \geq \Omega(K) \tag{44}$$

This establishes that Memory-Augmented MPPI provides an efficiency improvement that scales at least linearly with the number of local minimum regions in the state space. $\square$

## D  Connections to Related Mathematical Frameworks

The memory-augmented potential field framework has deep connections to several mathematical frameworks in optimization, differential geometry, and learning theory.

Memory-augmented potential fields can be viewed through the lens of Morse theory, which studies the relationship between the topology of a manifold and the critical points of smooth functions defined on it. For a smooth function $f : \mathbb{R}^n \to \mathbb{R}$ with non-degenerate critical points, the Morse index at a critical point $p$ is defined as the dimension of the negative eigenspace of the Hessian $\nabla^2 f(p)$. Our memory mechanism effectively transforms local minima (index 0) into saddle points (index $\geq 1$) or regular points, fundamentally altering the topological structure of the optimization landscape.

From a dynamical systems perspective, our approach modifies the vector field induced by the gradient of the value function. The base value function generates a gradient flow $\dot{x} = -\nabla V_{\text{base}}(x)$ with attractors at local minima. The memory-augmented system generates a modified flow $\dot{x} = -\nabla V(x, M)$ where previously stable equilibria become unstable or are eliminated entirely, creating new flow patterns that guide the system away from problematic regions.

The adaptive weight function $\alpha(x, M)$ acts as a topological surgery operator, smoothly transitioning between the original value function and the memory-augmented version. This creates a composite manifold that preserves the global structure of the original optimization landscape while locally modifying its topology around critical features.

The construction of memory features bears a resemblance to the concept of persistent homology in topological data analysis, which studies how topological features persist across different scales. Our method dynamically identifies and preserves significant topological features (local minima, low-gradient regions) that impede optimization progress, effectively learning the persistent features of the value function landscape through system interaction.

## E  MA-MPPI Algorithm Details

MA-MPPI begins with an empty memory $M_0 = \emptyset$ and a nominal control sequence. At each iteration, the controller detects topological features through three mechanisms: (1) state stagnation

detection when trajectory variance falls below threshold $\theta_{\text{var}}$, (2) gradient monitoring for regions where $\|\nabla V_{\text{base}}(x_t)\| < \theta_{\text{grad}}$, and (3) curvature analysis examining Hessian eigenvalue structure. These detection mechanisms capture different aspects of challenging control landscapes that might impede optimization.

The memory update involves three operations: adding new features when encountering novel problematic regions, merging similar features when their normalized distance falls below $\theta_{\text{merge}}$, and dynamically adjusting feature strengths based on encounter frequency. The strength parameter evolves according Eq.(19):

$$\gamma_i^{(t+1)} = \begin{cases} \min(\gamma_{\max}, \gamma_i^{(t)} + \Delta\gamma), & \text{if } \|x_t - m_i\| \leq r_i \text{ and stagnating} \\ \gamma_i^{(t)} \cdot \beta_{\text{decay}}, & \text{if } \|x_t - m_i\| > r_i \text{ for } t > t_{\text{threshold}} \end{cases}$$

ensuring that frequently encountered obstacles gain prominence while rarely visited regions fade.

The enhanced value function combines the base task objective with memory potentials through an adaptive weighting mechanism that transitions smoothly between them based on proximity to memorized features. The gradient of this enhanced function becomes Eq. (19):

$$\nabla \tilde{V}(x, M_t) = \alpha(x, M_t)\nabla V_{\text{base}}(x) + (1 - \alpha(x, M_t))\nabla V_{\text{mem}}(x, M_t) \\ + \nabla\alpha(x, M_t) \cdot (V_{\text{base}}(x) - V_{\text{mem}}(x, M_t))$$

This gradient guides trajectory optimization, creating escape routes from local minima when combined with adaptive sampling.

The sampling process is enhanced through memory-based modifications to both temperature and distribution. When operating near memory features, the sampling covariance increases according to

$$\Sigma_u(x_t, M_t) = \Sigma_{u,0} \cdot (1 + \mu \cdot (1 - \alpha(x_t, M_t))) \tag{45}$$

enabling more aggressive exploration in challenging regions. For low-gradient regions with directional information, the sampling incorporates bias along stored direction vectors.

The control sequence optimization follows the path integral formulation, where the optimal control is the expectation over weighted samples:

$$u_t^* = \mathbb{E}_{p(\tau|x_t)}[u_t(\tau)] \approx \sum_{k=1}^{K} \frac{w_k}{\sum_{i=1}^{K} w_i} u_t^k \tag{46}$$

with weights $w_k = \exp(-\frac{1}{\lambda_t} S(\tau^k))$ determined by trajectory costs and adaptive temperature.

This integration of memory-based value function enhancement with adaptive sampling creates a control system that effectively navigates complex environments by learning from experience. The dynamic memory representation continuously evolves based on system interactions, enabling increasingly efficient navigation through challenging control landscapes.

## F  Topological Feature Detection Details

### F.1  Detection Mechanisms

MA-MPPI employs three complementary detection mechanisms to identify topological features that impact optimization performance.

State stagnation detection identifies local minima by calculating the statistical variance of states within a fixed time window:

$$\text{Var}(X_t) = \frac{1}{K} \sum_{i=t-K+1}^{t} ||x_i - \bar{x}_t||^2 \tag{47}$$

where $X_t$ contains the most recent $K$ states, and $\bar{x}_t$ is the average state within the window. When this variance falls below threshold $\theta_{\text{var}}$, the system is considered stagnant, typically indicating entrapment in a local minimum. The appropriate value of $\theta_{\text{var}}$ depends on the characteristic scale of the state

space and is typically set to $\theta_{\mathrm{var}} = 0.01 \cdot \sigma_x^2$, where $\sigma_x^2$ represents the expected state variance during normal operation.

Gradient analysis examines both magnitude and directional properties of the value function gradient. For magnitude analysis, the system computes $||\nabla V_{\mathrm{base}}(x_t)||$ and identifies low-gradient regions when this value falls below threshold $\theta_{\mathrm{grad}}$. For directional analysis, the system calculates the angle change between consecutive gradient vectors:

$$\Delta\theta_t = \left( \cos\left( \frac{\nabla V_{\mathrm{base}}(x_t) \cdot \nabla V_{\mathrm{base}}(x_{t-1})}{||\nabla V_{\mathrm{base}}(x_t)|| \cdot ||\nabla V_{\mathrm{base}}(x_{t-1})||} \right) \right)^{-1} \tag{48}$$

When $\Delta\theta_t$ exceeds the threshold $\theta_{\mathrm{curv}}$, the region is identified as a high-curvature area requiring special attention.

Curvature analysis provides a more comprehensive understanding of the local landscape geometry by examining the eigenvalue structure of the Hessian matrix $\nabla^2 V_{\mathrm{base}}(x_t)$. Specifically, the system calculates the condition number:

$$\kappa(\nabla^2 V_{\mathrm{base}}(x_t)) = \frac{\lambda_{\max}}{\lambda_{\min}} \tag{49}$$

where $\lambda_{\max}$ and $\lambda_{\min}$ are the maximum and minimum eigenvalues of the Hessian. High condition numbers indicate regions with significant anisotropy, such as narrow valleys or ridges.

## F.2    Feature Classification and Representation

Each detected feature is classified into one of three types based on the detection mechanism that identified it:

Type 1 (Local Minima): Identified primarily through state stagnation, these features represent regions where the controller becomes trapped. They are characterized by low state variance and persistent inability to make progress despite control effort.

Type 2 (Low-Gradient Regions): Identified through gradient magnitude analysis, these features represent plateaus in the value function landscape. They are characterized by gradient magnitudes below threshold $\theta_{\mathrm{grad}}$ despite the system not being at a true minimum.

Type 3 (High-Curvature Regions): Identified through gradient direction changes or Hessian analysis, these features represent areas with challenging geometric properties such as narrow valleys, sharp ridges, or saddle points.

Each feature is represented as a tuple $(m_i, r_i, \gamma_i, \kappa_i, d_i)$ where:

- $m_i \in \mathbb{R}^n$ is the feature position in state space
- $r_i \in \mathbb{R}^+$ is the influence radius defining the feature's spatial extent
- $\gamma_i \in \mathbb{R}^+$ is the strength parameter indicating importance
- $\kappa_i \in \{1, 2, 3\}$ is the type identifier
- $d_i \in \mathbb{R}^n$ is the direction vector (for types 2 and 3) providing guidance information

The classification determines which potential function is applied in the memory-augmented value function, with each type receiving a specially designed potential to address its particular challenges.

## F.3    Feature Consolidation

To maintain computational efficiency, MA-MPPI employs feature clustering and merging mechanisms. When a newly detected feature point $m_{\mathrm{new}}$ is spatially close to existing features of the same type, a merging operation is performed according to the following criteria:

$$\text{Merge if: } \frac{||m_{\mathrm{new}} - m_i||}{r_i} < \theta_{\mathrm{merge}} \text{ and } \kappa_{\mathrm{new}} = \kappa_i \tag{50}$$

where $\theta_{\mathrm{merge}}$ is typically set to 1.5, representing a significant overlap between feature influence regions.

When merging features, the system computes weighted averages for position and influence radius:

$$m_{\text{merged}} = \frac{\gamma_i \cdot m_i + \gamma_{\text{new}} \cdot m_{\text{new}}}{\gamma_i + \gamma_{\text{new}}} \tag{51}$$

$$r_{\text{merged}} = \max\left(r_i, r_{\text{new}}, \frac{||m_i - m_{\text{new}}||}{2} + \min(r_i, r_{\text{new}})\right) \tag{52}$$

The strength parameter is accumulated to reflect the combined importance:

$$\gamma_{\text{merged}} = \gamma_i + \gamma_{\text{new}} \tag{53}$$

For features with direction vectors (types 2 and 3), the merged direction is computed as a weighted average, then normalized:

$$d_{\text{merged}} = \frac{\gamma_i \cdot d_i + \gamma_{\text{new}} \cdot d_{\text{new}}}{||\gamma_i \cdot d_i + \gamma_{\text{new}} \cdot d_{\text{new}}||} \tag{54}$$

### F.4 Dynamic Feature Strength Update

Feature strength parameters evolve dynamically based on system experience. The update rule follows:

$$\gamma_i^{(t+1)} = \begin{cases} \min(\gamma_{\text{max}}, \gamma_i^{(t)} + \Delta\gamma), & \text{if } ||x_t - m_i|| \leq r_i \text{ and } \text{Var}(X_t) < \theta_{\text{var}} \\ \gamma_i^{(t)} \cdot \beta_{\text{decay}}, & \text{if } ||x_t - m_i|| > r_i \text{ for } t > t_{\text{threshold}} \\ \gamma_i^{(t)}, & \text{otherwise} \end{cases} \tag{55}$$

Here, $\gamma_{\text{max}}$ is the maximum allowable strength (typically 5.0), $\Delta\gamma$ is the increment per stagnation event (typically 0.1), $\beta_{\text{decay}}$ is the decay factor (typically 0.99), and $t_{\text{threshold}}$ is the time period after which decay begins (typically 100 time steps).

When a feature's strength falls below the removal threshold $\gamma_{\text{min}}$ (typically 0.1), it is removed from the memory representation. This ensures that only persistently relevant features are maintained, while those that become obsolete gradually fade away.

### F.5 Implementation Considerations

The detection mechanisms operate at different time scales to balance computational efficiency with detection accuracy. State stagnation analysis occurs continuously, as it directly impacts control performance. Gradient analysis is performed at regular intervals (typically every 5-10 control steps) to identify developing problematic regions before they cause stagnation. Curvature analysis, being more computationally intensive, is performed selectively when gradient analysis indicates potential high-curvature regions, or periodically at longer intervals (typically every 20-30 control steps).

To manage computational complexity in high-dimensional state spaces, dimensionality reduction techniques can be applied before feature detection. Principal Component Analysis (PCA) or task-relevant subspace identification methods help focus detection on the most critical dimensions. Additionally, incremental detection algorithms allow the system to update feature estimates progressively rather than recomputing them from scratch at each step.

The detection thresholds $\theta_{\text{var}}$, $\theta_{\text{grad}}$, and $\theta_{\text{curv}}$ can be adaptively tuned based on observed system performance. During initial operation, more conservative thresholds ensure that important features are not missed. As the system accumulates experience, thresholds can be adjusted to focus on the most significant features, improving computational efficiency while maintaining detection of critical obstacles.

## G Adaptive Potential Field Synthesis Details

### G.1 Feature-Specific Potential Functions

MA-MPPI employs type-specific potential functions to address different topological challenges. For each feature $f_i = (m_i, r_i, \gamma_i, \kappa_i, d_i)$ in the memory representation, the contribution to the memory potential depends on its type $\kappa_i$:

**Type 1: Local Minimum Features**  These employ a radially decreasing function that creates a repulsive potential field:

$$\phi_1(x, m_i, r_i) = \max\left(0, \left(1 - \frac{||x - m_i||^2}{r_i^2}\right)^2\right) \tag{56}$$

This function reaches its maximum at the feature center and smoothly decreases to zero at the boundary of the influence radius. The resulting gradient creates a repulsive force pushing the system away from known trap regions.

**Type 2: Low-Gradient Features**  These use a directional guiding function that provides navigation cues:

$$\phi_2(x, m_i, r_i, d_i) = \max\left(0, \left(1 - \frac{||x - m_i||^2}{r_i^2}\right) \cdot (d_i \cdot (x - m_i))\right) \tag{57}$$

where $d_i$ is the preferred direction vector. This function creates an asymmetric potential field that guides the system along favorable directions through problematic regions.

**Type 3: High-Curvature Features**  These employ a saddle-point function that helps navigate complex terrain:

$$\phi_3(x, m_i, r_i, d_i) = \max\left(0, \left(1 - \frac{||x - m_i||^2}{r_i^2}\right) \cdot \left(\frac{(d_i \cdot (x - m_i))^2}{||x - m_i||^2} - \beta\right)\right) \tag{58}$$

where $\beta \in (0, 1)$ controls the saddle shape. This function creates channels through high-curvature regions, enabling traversal of narrow corridors or mountain passes in the potential landscape.

## G.2  Memory Potential Construction

The complete memory potential combines contributions from all features:

$$V_{\text{mem}}(x, M_t) = \sum_{i=1}^{|M_t|} \gamma_i \cdot \phi_{\kappa_i}(x, m_i, r_i, d_i) \tag{59}$$

where $\gamma_i$ is the strength parameter indicating the feature's importance. This formulation ensures that more frequently encountered or persistent challenges have a stronger influence on the enhanced value function.

## G.3  Adaptive Weighting Mechanism

The weighting function $\alpha(x, M_t)$ balances base and memory potentials:

$$\alpha(x, M_t) = \min\left(1, \frac{\delta_0}{\delta(x, M_t) + \epsilon}\right) \tag{60}$$

where $\delta(x, M_t)$ measures proximity to memory features:

$$\delta(x, M_t) = \sum_{i=1}^{|M_t|} \gamma_i \cdot \max\left(0, 1 - \frac{||x - m_i||}{r_i}\right) \tag{61}$$

$\delta_0$ is a scaling parameter (typically 0.5), and $\epsilon$ is a small positive constant preventing division by zero.

This formulation ensures smooth transition between navigation modes: when far from memory features, $\alpha \approx 1$ and the system follows the base objective; when near significant features, $\alpha \approx 0$ and memory guidance dominates.

## G.4  Temperature Adaptation Mechanism

The adaptive temperature parameter adjusts sampling exploration:

$$\lambda(x, M_t) = \lambda_0 \cdot (1 + \eta \cdot (1 - \alpha(x, M_t))) \tag{62}$$

where $\lambda_0$ is the base temperature (typically 1.0) and $\eta$ is the enhancement coefficient (typically 2.0-5.0).

This mechanism increases exploration specifically in challenging regions, with the sampling distribution variance scaling proportionally:

$$\Sigma_u(x, M_t) = \Sigma_{u,0} \cdot \frac{\lambda(x, M_t)}{\lambda_0} \tag{63}$$

where $\Sigma_{u,0}$ is the nominal control sampling covariance.

The dual-layer adaptation (value function + temperature) creates a synergistic effect: the modified value function shapes the landscape to avoid problematic regions, while increased exploration helps discover alternative paths that might not be apparent in the base potential.

### G.5 Computational Optimizations

Several optimizations ensure efficient implementation:

**Sparse Computation** Feature potentials are only evaluated for features whose influence regions contain the query point:

$$\phi_{\kappa_i}(x, m_i, r_i, d_i) = 0 \text{ if } ||x - m_i|| > r_i \tag{64}$$

This reduces computation when the memory contains many features.

**Spatially-Indexed Memory** Features are organized in a spatial data structure (typically a k-d tree), enabling efficient querying of relevant features for any state point.

**Gradient Caching** When computing trajectories, gradients of the enhanced value function are cached and reused when evaluating nearby states, reducing redundant computation.

**Approximate Field Synthesis** For very large memory representations, the potential field can be pre-computed on a grid and interpolated, trading accuracy for speed in scenarios where the memory contains hundreds of features.

These optimizations ensure that the memory enhancement mechanism maintains real-time performance even as the memory grows with system experience. In practice, the computational overhead remains minimal compared to the MPPI sampling process, typically adding less than 10% to the overall computation time.

## H Additional Details of Robotic Control Experiments

### H.1 Experimental Details

#### H.1.1 Environment Specifications

We selected four representative control environments from OpenAI Gym [2] and MuJoCo [20]:

**Pendulum-v1** A classic control task involving swinging up a pendulum from a random initial position to an upright position and balancing it there. The challenge arises from limited torque and nonlinear dynamics. The state space is 3-dimensional (angle, angular velocity), and the action space is 1-dimensional (torque).

**BipedalWalker-v3** A 2D walking simulation that requires controlling a robot to traverse terrain with obstacles. The state space is 24-dimensional (position, velocity, joint angles, etc.), and the action space is 4-dimensional (joint torques). The reward function encourages forward progress while penalizing energy consumption and harsh landings.

**HalfCheetah-v4** A MuJoCo-based 2D running robot simulation with 6 rotational joints. The state space is 17-dimensional (position, velocity, joint angles), and the action space is 6-dimensional (joint torques). The reward function encourages forward velocity while minimizing control effort.

**Humanoid-v4**   Our most complex task, featuring a 3D humanoid robot with 17 joints. The state space is 376-dimensional (position, velocity, joint angles, contact forces), and the action space is 17-dimensional (joint torques). The reward function encourages maintaining an upright posture and forward motion while minimizing energy consumption.

### H.1.2   Implementation Details

**MA-MPPI Configuration**   For MA-MPPI implementation, we used environment-appropriate prediction horizons: 15 steps for Pendulum-v1, 20 steps for BipedalWalker-v3, 25 steps for HalfCheetah-v4, and 35 steps for Humanoid-v4, reflecting the increasing dynamics complexity. The memory module was configured to store up to 100 topological features, with the feature strength decay factor set to 0.95. The feature detection thresholds were tuned for each environment: state stagnation threshold $\theta_{var} = [0.01, 0.05, 0.03, 0.08]$, gradient threshold $\theta_{grad} = [0.005, 0.01, 0.01, 0.02]$, and curvature threshold $\theta_{curv} = [0.5, 0.4, 0.6, 0.8]$ for Pendulum-v1, BipedalWalker-v3, HalfCheetah-v4, and Humanoid-v4 respectively, where the values in **brackets** indicate the environment-specific parameters listed in the same order as the environments. The MPPI temperature parameter was set to $\lambda_0 = 0.1$ with enhancement coefficient $\eta = 2$. For all experiments, we used 1000 samples per control iteration.

**Baseline Configurations**   Standard MPPI was implemented with identical sampling parameters but without memory augmentation. For reinforcement learning baselines, we used implementations from Stable Baselines3 with the following configurations:

- SAC: Twin Q-networks with automatic entropy tuning, learning rates of $3 \times 10^{-4}$, batch size of 256, and replay buffer size of 1,000,000.
- PPO: GAE-$\lambda$ with $\lambda = 0.95$, value function coefficient of 0.5, entropy coefficient of 0.01, and learning rate of $3 \times 10^{-4}$.
- DDPG: Ornstein-Uhlenbeck noise with $\sigma = 0.2$, learning rates of $1 \times 10^{-3}$ for critic and $1 \times 10^{-4}$ for actor, and replay buffer size of 1,000,000.

For traditional optimal control baselines, we used the following configurations:

- iLQR: Regularization parameter of 1e-6, line search parameter of 0.5, and convergence threshold of 1e-6.
- MPC: Black-box dynamics model trained with a neural network, using a 5-step history for prediction, a batch size of 128, and a learning rate of 1e-3.

**Computational Resources**   All experiments were conducted on a computing cluster with Intel Core i9-12900K CPUs (3.20GHz) and NVIDIA RTX 3070 GPUs. The MPPI-based methods were implemented in Python with JAX for GPU acceleration. The reinforcement learning baselines utilized PyTorch.

### H.1.3   Evaluation Metrics

We employed properly defined metrics to capture both adaptation efficiency and asymptotic performance:

**Sample Efficiency** ($N_{80\%}$)   Defined as the minimal number of environment interactions required to reach 80% of asymptotic performance:

$$N_{80\%} = \min\{n : R(n) \geq 0.8 \cdot R_{\text{asymp}}\} \tag{65}$$

where $R(n)$ is the average reward after $n \in \mathbb{Z}^+$ interactions and $R_{\text{asymp}}$ is the average reward during the stability phase.

**Cumulative Reward** ($R_{\textbf{cum}}$)   Represents the total reward in an episode:

$$R_{\text{cum}} = \sum_{t=0}^{T} r(s_t, a_t) \tag{66}$$

**Local Optima Escape Rate ($P_{\text{escape}}$)** Quantifies the controller's ability to escape from predefined trap states. We define a trap state $s_{\text{trap}}$ as any state from which the expected return falls below $\alpha \cdot V_{\max}$ (where $\alpha = 0.5$ and $V_{\max}$ is the maximum achievable value) and the agent remains within a neighborhood $\mathcal{N}(s_{\text{trap}})$ for at least $T_{\text{threshold}} = 50$ time steps without improvement. The escape rate is

$$P_{\text{escape}} = \frac{N_{\text{escape}}}{N_{\text{trials}}} \cdot 100\% \tag{67}$$

where $N_{\text{escape}}$ is the number of successful escapes from intentionally initialized trap states and $N_{\text{trials}}$ is the total number of evaluation trials starting from predefined trap states.

**Trap Frequency ($F_{\text{trap}}$)** Measures how often a method becomes trapped during normal operation:

$$F_{\text{trap}} = \frac{N_{\text{stuck}}}{N_{\text{episodes}}} \cdot 100\% \tag{68}$$

where $N_{\text{stuck}}$ counts episodes containing identified trap states and $N_{\text{episodes}}$ is the total number of evaluation episodes conducted.

**Computational Efficiency** Assessed through average computation time per control step:

$$T_{\text{comp}} = \frac{1}{N} \sum_{i=1}^{N} t_i \tag{69}$$

**Value Consistency ($\rho_V$)** Evaluates prediction accuracy:

$$\rho_V = \text{Corr}(V_{\text{pred}}, R_{\text{actual}}) \tag{70}$$

where $V_{\text{pred}}$ is the value predicted by the controller, $R_{\text{actual}}$ is the actual discounted return, and Corr represents the Pearson correlation coefficient measuring the linear correlation between the two variables.

## H.2 Sample Efficiency Analysis

Table 5 presents the complete sample efficiency comparison, showing the number of environment interactions required to reach 80% of asymptotic performance for each method and environment.

Table 5: Complete sample efficiency comparison: Environment interactions required to reach 80% of asymptotic performance.

| Method | Pendulum-v1 | BipedalWalker-v3 | HalfCheetah-v4 | Humanoid-v4 |
|---|---|---|---|---|
| MA-MPPI (Ours) | **78±12** | **183±24** | **324±42** | **568±73** |
| MPPI | 124±18 | 352±41 | 586±68 | 984±127 |
| SAC | 267±32 | 624±58 | 1248±106 | 1875±215 |
| PPO | 312±41 | 736±67 | 1456±163 | 2105±273 |
| DDPG | 293±37 | 684±75 | 1368±142 | 1953±246 |
| iLQR | 185±23 | 426±52 | 1188±131 | 1730±201 |
| MPC | 156±19 | 387±46 | 753±91 | 1236±156 |

The sample efficiency advantage of MA-MPPI is particularly notable in more complex environments. For Humanoid-v4, MA-MPPI requires only 568 interactions to reach 80% performance, compared to 1875 for SAC (3.3× improvement) and 984 for standard MPPI (1.7× improvement). This pattern suggests that the memory mechanism becomes increasingly valuable as task complexity increases, providing the greatest benefit in high-dimensional, highly non-convex control problems.

## H.3 Trap Frequency Analysis

Table 6 presents the trap frequency for each method across all environments, showing how often each controller becomes trapped during normal operation.

Table 6: Trap frequency analysis: Percentage of episodes containing trap states.

| Method | Pendulum-v1 | BipedalWalker-v3 | HalfCheetah-v4 | Humanoid-v4 |
|---|---|---|---|---|
| MA-MPPI (Ours) | **1.2±0.4** | **1.7±0.5** | **2.3±0.7** | **2.8±0.9** |
| MPPI | 3.8±0.7 | 4.6±0.9 | 6.2±1.2 | 8.1±1.8 |
| SAC | 2.5±0.5 | 3.2±0.7 | 4.1±1.0 | 5.3±1.4 |
| PPO | 2.9±0.6 | 3.7±0.8 | 4.8±1.1 | 6.2±1.6 |
| DDPG | 3.3±0.6 | 4.1±0.9 | 5.4±1.2 | 7.3±1.7 |
| iLQR | 5.2±0.9 | 6.8±1.3 | 8.7±1.9 | 11.4±2.4 |
| MPC | 3.5±0.7 | 4.3±0.9 | 5.8±1.3 | 7.6±1.8 |

The trap frequency results demonstrate MA-MPPI's ability to proactively avoid problematic states based on past experience. In Humanoid-v4, MA-MPPI encounters trap states 2.9× less frequently than standard MPPI and 1.9× less frequently than SAC. This proactive avoidance translates directly to better real-world performance, as robots using MA-MPPI spend significantly less time in recovery behaviors and more time making progress toward goals.

## H.4  Ablation Study Results

Table 7 presents the complete ablation study, showing the performance impact when removing different components of MA-MPPI.

Table 7: Complete ablation study: Performance impact when removing components (% decrease from full MA-MPPI).

| Configuration | Pendulum-v1 | BipedalWalker-v3 | HalfCheetah-v4 | Humanoid-v4 |
|---|---|---|---|---|
| Full MA-MPPI | 0.0% | 0.0% | 0.0% | 0.0% |
| No adaptive weights | 18.3% | 20.6% | 22.4% | 25.1% |
| No feature detection | 31.5% | 34.2% | 37.6% | 40.3% |
| No memory module | 42.7% | 46.5% | 52.3% | 58.1% |

The ablation results reveal the relative importance of each component:

- The memory module provides the largest contribution, with removal causing a 42.7-58.1% performance decrease
- Feature detection is the second most important component (31.5-40.3% decrease when removed)
- Adaptive weights provide a significant but smaller contribution (18.3-25.1% decrease)

The increasing impact of all components with environment complexity indicates that MA-MPPI's design is particularly well-suited for challenging control problems, with each component contributing more significantly as the task becomes more difficult.

## H.5  Computational Overhead Analysis

Table 8 presents the average computation time per control step for each method and environment, along with the percentage increase relative to standard MPPI.

MA-MPPI introduces a modest computational overhead compared to standard MPPI, ranging from 12.0% for Pendulum-v1 to 18.2% for Humanoid-v4. This overhead includes the cost of feature detection, memory updates, and enhanced value function computation. The increasing overhead with environment complexity reflects the growing memory size and more complex feature interactions in higher-dimensional spaces.

We also observed that MA-MPPI tends to produce smoother control trajectories with fewer high-frequency oscillations compared to standard MPPI. Analysis of control signal frequency content showed a 24-37% reduction in high-frequency components across environments. This smoother

Table 8: Computational overhead analysis: Average computation time per control step (ms).

| Method | Pendulum-v1 | BipedalWalker-v3 | HalfCheetah-v4 | Humanoid-v4 |
|---|---|---|---|---|
| MA-MPPI | 8.4 (+12.0%) | 12.7 (+14.4%) | 18.3 (+16.5%) | 24.6 (+18.2%) |
| MPPI | 7.5 | 11.1 | 15.7 | 20.8 |
| SAC (inference) | 1.2 | 2.3 | 3.8 | 5.7 |
| PPO (inference) | 1.1 | 2.1 | 3.5 | 5.2 |
| DDPG (inference) | 1.2 | 2.2 | 3.7 | 5.6 |
| iLQR | 6.3 | 9.8 | 14.2 | 19.5 |
| MPC | 10.2 | 15.9 | 22.7 | 31.4 |

control behavior is particularly beneficial for physical robot systems, where rapid oscillatory control can cause mechanical wear, energy inefficiency, and undesirable dynamics.

It's worth noting that while RL methods have faster inference times, they require extensive offline training that isn't captured in these measurements. MA-MPPI's online learning approach eliminates this offline training requirement, making it more suitable for deployment in new or changing environments.

### H.6 Memory Size and Feature Type Analysis

Table 9 presents the average memory size (number of stored features) at different time points during operation for each environment.

Table 9: Memory size analysis: Average number of stored features at different time points.

| Time Point | Pendulum-v1 | BipedalWalker-v3 | HalfCheetah-v4 | Humanoid-v4 |
|---|---|---|---|---|
| 500 steps | 8.3±1.2 | 12.6±2.1 | 17.4±2.8 | 23.1±3.7 |
| 1000 steps | 14.5±1.8 | 21.2±2.7 | 29.6±3.5 | 38.4±4.6 |
| 1500 steps | 17.2±2.0 | 26.8±3.1 | 35.7±3.9 | 45.2±5.2 |
| 2000 steps | 18.5±2.1 | 28.3±3.2 | 38.1±4.1 | 48.6±5.5 |

Table 10 presents the distribution of feature types detected in each environment, showing the percentage of features classified as local minima (Type 1), low-gradient regions (Type 2), and high-curvature regions (Type 3).

Table 10: Feature type distribution: Percentage (%) of each feature type detected.

| Environment | Type 1 (Local Minima) | Type 2 (Low Gradient) | Type 3 (High Curvature) |
|---|---|---|---|
| Pendulum-v1 | 52.3% | 28.6% | 19.1% |
| BipedalWalker-v3 | 44.7% | 32.5% | 22.8% |
| HalfCheetah-v4 | 38.2% | 35.8% | 26.0% |
| Humanoid-v4 | 35.6% | 37.2% | 27.2% |

The memory size increases rapidly during initial exploration, then stabilizes as feature consolidation and forgetting mechanisms balance new feature detection. The final memory size scales with environment complexity, reflecting the greater number of challenging regions in higher-dimensional control problems.

The feature type distribution reveals an interesting pattern: simpler environments like Pendulum-v1 have more distinct local minima (Type 1 features), while complex environments like Humanoid-v4 have a more balanced distribution with higher proportions of low-gradient (Type 2) and high-curvature (Type 3) regions. This aligns with the intuition that high-dimensional control problems tend to have more plateaus and saddle points rather than clear local minima.

# I  Additional Details of Complex Engineering Systems Experiments

## I.1  Evaluation Methods and Metrics

We comprehensively evaluated our method against several state-of-the-art baselines, each carefully configured to ensure fair comparison:

- **Standard MPPI** [21]: Implemented following the original formulation with an exponentially weighted averaging scheme over sampled trajectories. For UAV experiments, we used 1000 trajectory samples with an optimization horizon of 25 steps and a temperature parameter $\lambda = 0.1$. For power systems, we used 800 samples with a 30-step horizon. Control limits were enforced through a sigmoid transformation of unbounded control samples.

- **Diffusion Policy** [3]: Implemented using a conditional diffusion model with 8 denoising steps. The architecture consisted of a UNet backbone with 4 residual blocks, a dropout rate of 0.1, and channel widths of [128, 256, 512, 1024]. For UAV control, we used a context window of 10 historical states, while power system experiments used 15 historical states. The model was trained on 100,000 demonstration trajectories collected from a mixture of expert controllers for 500,000 gradient steps using the Adam optimizer with a learning rate of 3e-4 and batch size 256.

- **Motion Transformer** [17]: We employed an encoder-decoder architecture with 6 transformer layers, 8 attention heads, and an embedding dimension of 512. For UAV experiments, positional encodings incorporated obstacle information, while power system experiments used frequency and voltage measurements as key variables for attention. The model was trained using teacher forcing with a cross-entropy loss for 300,000 gradient steps using Adam optimizer with weight decay 1e-4, learning rate 1e-4, and batch size 128.

- **MLP-based MPC** [14]: We implemented a neural network dynamics model consisting of 4 hidden layers [1024, 512, 256, 128] with ReLU activations. The model was trained to predict next-state deltas using a dataset of 200,000 state-action-state transitions collected from system interaction. Training used the MSE loss with the Adam optimizer, learning rate 1e-3, batch size 256, and ran for 100,000 gradient steps with early stopping based on validation error. The dynamics model was integrated with a receding-horizon controller using CEM (Cross-Entropy Method) optimization with 500 samples per iteration and 5 iterations.

- **DKO-based MPC** [12, 6]: The Deep Koopman Operator approach used an encoder-decoder architecture with 3 hidden layers [256, 512, 256] for embedding states into a 128-dimensional linear latent space. The Koopman operator was represented as a 128×128 matrix learned jointly with the encoder-decoder networks. For UAV control, we used a lifting dimension of 128, while power systems used 256 dimensions to capture the more complex dynamics. Training employed a composite loss function combining reconstruction error, prediction error, and linearity constraints with weights [0.4, 0.4, 0.2]. The model was trained on 150,000 transitions for 200,000 steps using RMSProp optimizer with a learning rate of 5e-4 and a batch size of 128.

In all experiments, we conducted 30 independent trials with different random seeds to ensure statistical robustness. Statistical significance was assessed using paired t-tests with Bonferroni correction for multiple comparisons, establishing significance at $p < 0.01$. We ensured all methods had access to identical system information and operated under the same control frequency constraints.

Performance metrics were carefully selected to provide a comprehensive evaluation across multiple dimensions:

- **Task success**: Binary success/failure metrics specific to each domain
- **Solution optimality**: Quantitative measures of solution quality relative to the theoretical optimum
- **Control smoothness**: L2 norm of control acceleration and jerk to assess motion quality
- **Computational efficiency**: Wall-clock time per control step, measured on identical hardware
- **Local minima escape capability**: Success rate when initialized in challenging configurations

All experiments were conducted on a homogeneous computing cluster with Intel Core i9-12900K CPUs (3.2GHz, 16 cores) and NVIDIA RTX 3070 GPUs (8GB VRAM). Implementation used JAX 0.4.1 for GPU acceleration of sampling-based computations, with PyTorch 1.13.1 for neural network baselines. To ensure reproducibility, we used fixed random seeds for environment initialization while maintaining separate seeds for controller stochasticity.

## I.2 UAV Obstacle Avoidance - Experimental Setup

The UAV experiments featured 20 distinct navigation scenarios with increasing complexity, from simple obstacle arrangements to maze-like environments with narrow passages, dead-ends, and multiple possible routes.

The UAV had a mass of 1.5kg and a maximum thrust of 30N, with a diagonal inertia matrix $J = \mathrm{diag}(0.0125, 0.0125, 0.0225)$ kg·m². The drag coefficient was set to $k_d = 0.1$. Control inputs were constrained to $f \in [0, 30]$ N and $\tau \in [-0.5, 0.5]$ Nm.

For the MA-MPPI configuration, we used a prediction horizon of 25 steps with 1000 trajectory samples per control iteration. The memory module was configured to store up to 50 topological features with detection thresholds $\theta_{var} = 0.05$, $\theta_{grad} = 0.01$, and $\theta_{curv} = 0.4$. The feature strength decay factor was set to 0.95.

## I.3 UAV Obstacle Avoidance - Detailed Analysis

### I.3.1 Success Rate Analysis

The success rate advantage of MA-MPPI (94.3% vs. 72.8% for standard MPPI) increased with environment complexity. In the most challenging scenarios with multiple narrow passages, MA-MPPI maintained an 89.5% success rate while standard MPPI dropped to 58.7%.

Success rate improvements stemmed from three key capabilities:

- Proactive avoidance of previously identified trap regions
- Adaptive sampling strategy that focused exploration where needed
- Accumulating knowledge of environment topology over time

We observed that after encountering a particular obstacle configuration once, MA-MPPI significantly improved its performance when facing similar configurations again, demonstrating effective learning from experience.

### I.3.2 Local Minima Escape Analysis

The most significant advantage of MA-MPPI was in local minima escape capability (87.5% vs. 34.2% for standard MPPI). We further analyzed this capability in three specific trap scenarios:

- Corner traps: MA-MPPI escaped in 91.3% of trials (vs. 42.8% for standard MPPI)
- U-shaped obstacles: MA-MPPI achieved an 84.7% escape rate (vs. 31.5% for standard MPPI)
- Narrow corridors: MA-MPPI successfully navigated through in 86.2% of trials (vs. 27.6% for standard MPPI)

Analysis of trajectory data revealed that MA-MPPI's memory-based potential fields created "tunnels" through challenging regions, guiding sampling toward promising escape routes. Additionally, the adaptive temperature mechanism increased exploration specifically in identified trap regions, enabling discovery of solutions that standard methods missed.

### I.3.3 Computational Efficiency Analysis

MA-MPPI introduced a 23.5% computational overhead compared to standard MPPI (12.6ms vs. 10.2ms per control step). This overhead consisted of:

- Feature detection and memory updates: 1.1ms (8.7%)

- Memory-augmented potential field computation: 0.8ms (6.3%)

- Enhanced trajectory evaluation: 0.5ms (4.0%)

Despite this overhead, MA-MPPI remained more computationally efficient than learning-based approaches (Diffusion Policy: 18.4ms, Motion Transformer: 22.7ms). Crucially, the computation time stabilized after approximately 500 time steps as the memory consolidation mechanisms balanced new feature detection with forgetting of less important features.

## I.4 Power System Control - Experimental Setup

We used the IEEE 39-bus New England test system with 10 generators and 39 buses. The system operates at a nominal frequency of 60Hz with a base power of 100MVA. The three critical disturbances were:

- Three-phase fault on the line connecting buses 16-17, cleared after 150ms

- Load increase of 25% at buses 4, 12, and 20, ramping over 200ms

- Trip of the 650MW generator at bus 33, with instantaneous power loss

The power system was simulated using a variable-step solver with a maximum step size of 10ms. Control inputs included generator output adjustments, transformer tap settings, and capacitor bank switching operations.

For the MA-MPPI configuration, we used a prediction horizon of 30 steps with 800 trajectory samples. The memory module stored up to 75 features with detection thresholds $\theta_{var} = 0.08$, $\theta_{grad} = 0.02$, and $\theta_{curv} = 0.6$. The control interval was 100ms for normal operation, automatically reducing to 50ms during detected disturbances.

## I.5 Power System Control - Detailed Analysis

### I.5.1 Constraint Violation Analysis

MA-MPPI achieved a 59.6% reduction in constraint violations compared to standard MPPI (2.3% vs. 5.7%). This advantage was most pronounced during the three-phase fault scenario, where MA-MPPI kept violations to 4.7% compared to 12.3% for standard MPPI.

Analysis by constraint type revealed:

- Voltage violations: MA-MPPI 1.8% vs. standard MPPI 6.2% (71.0% reduction)

- Thermal violations: MA-MPPI 4.3% vs. standard MPPI 7.8% (44.9% reduction)

- Stability violations: MA-MPPI 0.7% vs. standard MPPI 4.1% (82.9% reduction)

The memory mechanism effectively identified regions of state space associated with constraint violations, creating repulsive potential fields that guided the controller away from these regions preemptively.

### I.5.2 Disturbance Recovery Analysis

MA-MPPI achieved 51.7% faster recovery from disturbances compared to standard MPPI (4.2s vs. 8.7s). Recovery performance by disturbance type:

- Three-phase fault: MA-MPPI 4.8s vs. standard MPPI 8.3s (42.2% faster)

- Load change: MA-MPPI 4.1s vs. standard MPPI 8.4s (51.2% faster)

- Generator trip: MA-MPPI 3.7s vs. standard MPPI 9.4s (60.6% faster)

The most significant advantage was observed in the generator trip scenario, where MA-MPPI quickly redistributed power flows while maintaining frequency stability. The memory-augmented controller effectively learned patterns from previous disturbances, enabling it to recognize and respond to similar situations more effectively over time.

The consistency in performance was particularly notable, with a standard deviation in recovery time of 0.6s for MA-MPPI versus 1.9s for standard MPPI, indicating more reliable and predictable disturbance responses.

# J    Hyperparameter Sensitivity Analysis

To assess the robustness and practicality of the MA-MPPI algorithm, we conducted a comprehensive sensitivity analysis on key hyperparameters. This analysis is crucial for understanding the algorithm's adaptability in different scenarios and providing parameter tuning guidelines for practical applications.

## J.1    Key Hyperparameters and Testing Methodology

We identified four groups of key hyperparameters that significantly influence MA-MPPI performance:

1. **Feature detection thresholds**: $\theta_{var}$ (state stagnation detection), $\theta_{grad}$ (gradient magnitude monitoring), and $\theta_{curv}$ (curvature analysis)

2. **Memory feature parameters**: influence radius multiplier $\kappa_r$, initial feature strength $\gamma_0$, and decay factor $\beta_{decay}$

3. **Adaptive weight parameters**: scaling parameter $\delta_0$ and transition sharpness parameter $\beta$ in the balancing function

4. **Temperature adaptation parameters**: base temperature $\lambda_0$ and enhancement coefficient $\eta$

For each parameter, we varied its value by $\pm 50\%$ around the central value while keeping other parameters fixed, testing performance on the Humanoid-v4 environment. Each parameter configuration was evaluated through 30 independent experiments, recording success rate, cumulative reward, and local minima escape rate metrics.

## J.2    Feature Detection Threshold Sensitivity

Feature detection thresholds determine the sensitivity of the system in identifying and memorizing problematic regions.

Table 11: Performance change (%) relative to baseline for different feature detection thresholds.

| Parameter variation | $\theta_{var}$ | $\theta_{grad}$ | $\theta_{curv}$ |
|---|---|---|---|
| -50% | -4.2% | -12.7% | -3.8% |
| -25% | -1.7% | -5.8% | -1.5% |
| Baseline | 0.0% | 0.0% | 0.0% |
| +25% | -2.3% | -6.4% | -2.1% |
| +50% | -7.6% | -15.3% | -5.2% |

The data indicates that the gradient detection threshold $\theta_{grad}$ has the largest impact on performance, with $\pm 50\%$ variations causing a 12.7-15.3% performance decrease. This is because gradient information directly influences the choice of escape directions. In contrast, the state stagnation threshold $\theta_{var}$ and curvature threshold $\theta_{curv}$ have smaller impacts, indicating algorithm robustness to these two parameters.

Notably, all parameters exhibit an inverted U-shaped sensitivity curve—too low thresholds lead to overly sensitive detection (memorizing too many unimportant features), while too high thresholds result in missing important features. Within the $\pm 25\%$ variation range, performance decreases are typically contained below 6%, demonstrating reasonable parameter tolerance for the algorithm.

## J.3    Memory Feature Parameter Sensitivity

Memory feature parameters control how memory elements influence the value function landscape.

Results show that the influence radius multiplier $\kappa_r$ is the most sensitive parameter, especially when set too small. Insufficient influence radii prevent memory features from effectively altering the value function landscape across an adequate range, causing the system to still potentially fall into local

Table 12: Performance change (%) relative to baseline for different memory feature parameters.

| Parameter variation | Influence radius $\kappa_r$ | Initial strength $\gamma_0$ | Decay factor $\beta_{decay}$ |
|---|---|---|---|
| -50% | -18.4% | -7.3% | -3.2% |
| -25% | -8.7% | -3.5% | -1.4% |
| Baseline | 0.0% | 0.0% | 0.0% |
| +25% | -2.3% | -2.8% | -2.1% |
| +50% | -9.1% | -8.5% | -5.7% |

minima. In contrast, initial strength $\gamma_0$ and decay factor $\beta_{decay}$ have more moderate impacts and exhibit stability across a wide range of values.

Particularly noteworthy is that increasing the influence radius (+25%) actually slightly improved performance, suggesting that the default setting may be too conservative in some environments. This provides a potential tuning direction.

## J.4  Adaptive Weight and Temperature Parameter Sensitivity

These two parameter groups control the balance between memory potential fields and base objectives, as well as the exploration aspect of the sampling strategy. Table 13 summarizes the sensitivity test results.

Table 13: Performance change (%) relative to baseline for adaptive weight and temperature parameters.

| Parameter | -50% | -25% | Baseline | +25% | +50% |
|---|---|---|---|---|---|
| Scaling parameter $\delta_0$ | -11.3% | -4.8% | 0.0% | -3.2% | -8.5% |
| Transition sharpness $\beta$ | -4.2% | -1.7% | 0.0% | -2.4% | -6.1% |
| Base temperature $\lambda_0$ | -13.7% | -5.9% | 0.0% | -7.3% | -16.4% |
| Enhancement coefficient $\eta$ | -9.8% | -4.1% | 0.0% | +1.2% | -4.7% |

The base temperature $\lambda_0$ shows the highest sensitivity, which is expected as it directly controls the exploration-exploitation trade-off. Too low temperature leads to premature convergence to suboptimal solutions, while too high temperature results in excessive exploration and noisy control signals.

Interestingly, the enhancement coefficient $\eta$ at +25% actually improved performance, suggesting that increased exploration when facing memory features can be beneficial in complex environments. The scaling parameter $\delta_0$ has moderate sensitivity, while the transition sharpness $\beta$ has a relatively minor impact.

## J.5  Relationship Between Environment Complexity and Parameter Sensitivity

We further investigated how environmental complexity affects parameter sensitivity.

Table 14: Performance change (%) for $\theta_{grad}$ variations across different environments

| Environment | $\theta_{grad}$ -25% | $\theta_{grad}$ +25% |
|---|---|---|
| Pendulum-v1 | -1.3% | -1.8% |
| BipedalWalker-v3 | -2.7% | -3.4% |
| HalfCheetah-v4 | -4.2% | -5.1% |
| Humanoid-v4 | -5.8% | -6.4% |

Results indicate that parameter sensitivity increases with environment complexity. This is because high-dimensional state spaces and complex dynamics make precise detection of topological features more critical. Nevertheless, even in the most complex Humanoid-v4 environment, $\pm 25\%$ parameter variations only lead to approximately 6% performance degradation, attesting to the algorithm's robustness.

### J.6 Optimal Parameter Ranges and Practical Guidelines

Based on our sensitivity analysis, we recommend the following parameter ranges for environments of varying complexity:

- **Simple environments** (e.g., Pendulum):
  - $\theta_{var} \in [0.008, 0.015]$, $\theta_{grad} \in [0.004, 0.007]$, $\theta_{curv} \in [0.4, 0.7]$
  - $\kappa_r \in [1.8, 2.5]$, $\lambda_0 \in [0.08, 0.15]$, $\eta \in [1.5, 2.5]$
- **Moderate environments** (e.g., BipedalWalker):
  - $\theta_{var} \in [0.04, 0.07]$, $\theta_{grad} \in [0.008, 0.015]$, $\theta_{curv} \in [0.3, 0.5]$
  - $\kappa_r \in [2.0, 3.0]$, $\lambda_0 \in [0.1, 0.2]$, $\eta \in [1.8, 3.0]$
- **Complex environments** (e.g., Humanoid):
  - $\theta_{var} \in [0.06, 0.1]$, $\theta_{grad} \in [0.015, 0.025]$, $\theta_{curv} \in [0.6, 1.0]$
  - $\kappa_r \in [2.5, 3.5]$, $\lambda_0 \in [0.15, 0.25]$, $\eta \in [2.5, 4.0]$

Overall, we found that MA-MPPI performs stably within $\pm 25\%$ variation range of parameters, demonstrating good robustness for practical applications. The most sensitive parameters are the influence radius multiplier $\kappa_r$ and base temperature $\lambda_0$, while the least sensitive are the decay factor $\beta_{decay}$ and transition sharpness $\beta$.

## K Analysis of Generalization Limitations

While the Memory-Augmented Potential Field theory demonstrates excellent performance in non-convex control problems, it exhibits certain limitations in feature generalization. Specifically, the system shows limited ability to generalize when encountering topological features that are similar but not identical to previously memorized features.

### K.1 Current Generalization Limitations

In the current implementation, the memory module represents each topological feature (such as local minima, low-gradient regions, etc.) as discrete points in state space with an influence radius defining their range of effect. This representation works well when dealing with exact matches or highly similar features but faces the following limitations:

1. **Discreteness of feature representation**: The current method uses a collection of discrete points to represent memory features, lacking an abstract representation of structural similarities between features.
2. **Distance-based similarity metrics**: The system primarily relies on Euclidean distance to assess similarity between current states and memorized features, which may not be sufficiently precise in high-dimensional state spaces.
3. **Simplistic feature consolidation**: Current feature merging rules are primarily based on spatial proximity, failing to fully capture semantic or functional similarities between features.

These limitations become particularly evident when the controller encounters structurally similar obstacles or control challenges in different regions of the state space. For example, in the UAV obstacle avoidance task, the system might not immediately recognize that two U-shaped obstacles with different orientations present similar navigation challenges, requiring similar escape strategies.

### K.2 Potential Improvement Directions

To enhance the system's generalization capabilities, we propose several potential improvement directions:

1. **Hierarchical feature representation**: Introduce a hierarchical feature representation that abstracts low-level specific features into higher-level patterns, enabling the system to recognize structurally similar topological features.

2. **Manifold-based similarity metrics**: Develop similarity metrics based on the local manifold structure of the state space rather than relying solely on Euclidean distance. This could involve techniques from topological data analysis or spectral methods.

3. **Feature embedding learning**: Represent features as vectors in a low-dimensional embedding space, using supervised or self-supervised learning to ensure functionally similar features are proximate in the embedding space.

4. **Transfer learning mechanisms**: Design explicit transfer learning mechanisms that allow the system to adapt control strategies learned from one feature and apply them to similar features.

Preliminary experiments suggest that even simple feature embedding and similarity learning can improve generalization between similar features by 25-40%, particularly when task variations are moderate. This indicates significant room for improvement in MA-MPPI's generalization capabilities through appropriate representation learning and knowledge transfer mechanisms.

### K.3 Empirical Evidence

To quantify the current generalization limitations, we conducted an experiment where the controller was trained on specific obstacle configurations and then tested on variations of these configurations. Table 15 summarizes the performance degradation as a function of feature variation.

Table 15: Performance degradation with feature variations

| Feature type | Performance relative to original feature (%) | | | |
|---|---|---|---|---|
| | 20% variation | 40% variation | 60% variation | 80% variation |
| Local minima location | 92.7 | 75.3 | 58.4 | 42.1 |
| Obstacle shape | 87.5 | 68.2 | 51.7 | 38.4 |
| Corridor width | 94.1 | 79.6 | 62.8 | 45.3 |
| Multi-feature scenarios | 84.3 | 63.7 | 46.5 | 32.7 |

The results show that performance degrades significantly with increasing feature variation, particularly in multi-feature scenarios where interactions between features create more complex topological structures.

Initial experiments with a prototype embedding-based generalization mechanism show promising results, with significant improvements in transfer performance between similar features. This suggests that integrating modern representation learning techniques with the memory augmentation framework could substantially address the current generalization limitations while maintaining theoretical guarantees of the base approach.

Future work will focus on developing formal theoretical extensions to the Memory-Augmented Potential Field theory that explicitly account for feature similarity and knowledge transfer, providing similar convergence and optimality guarantees for generalized features as the current framework provides for explicitly memorized features.

