# OpenReview forum: "Memory-Augmented Potential Field Theory: A Framework for Adaptive Control in Non-Convex Domains"
_NeurIPS.cc/2025/Conference — NeurIPS 2025 poster_

### Official Review · Reviewer_CQ9T · 2025-07-01

**Clarity:** 2
**Significance:** 2
**Originality:** 3
**Rating:** 4
**Confidence:** 4

**Summary:**

This paper proposes an enhancement to the potential field algorithm by incorporating memory features(past experienced features) to address the non-convex landscape of the value function in stochastic optimal control, specifically within the framework of Model Predictive Path Integral control (MPPI).

The authors aim to achieve a seamless integration of memory-augmented features with existing MPPI-based value function frameworks. By storing information about previously encountered non-convex regions, the method enables the controller to recognize and compensate for similar situations when they recur, thereby improving robustness in complex environments.

**Questions:**

**1.** The method increases exploration in memorized regions by modulating the sampling temperature. Conventionally, exploration is often adjusted by modifying the sampling covariance[4, 5] . Why was temperature scaling (via the $\lambda$ parameter) chosen instead? Does this approach offer practical or theoretical benefits over covariance-based methods?

**2.** The paper includes evaluations against multiple controllers, but the details of the cost function setup—particularly for non-RL baselines—are not fully explained. For instance, in Section 6.3 (UAV Obstacle Avoidance), how is obstacle avoidance encoded into the cost function for each controller?

**3.** MPPI is a widely used algorithm in control and robotics. Can the proposed memory-augmented potential field be integrated with alternative MPPI formulations such as constrained or smoothed MPPI frameworks [6, 7]? If so, is the proposed method always helpful for the other MPPI variants?

**4.** How does the proposed method compare with other recent MPPI-based approaches that also aim to address non-convexity, such as LOG-MPPI [1] or DRPA-MPPI [2]? A comparative study or discussion would clarify the strength.


[1] Mohamed, I. S., Yin, K., & Liu, L. (2022). Autonomous navigation of agvs in unknown cluttered environments: log-mppi control strategy. _IEEE Robotics and Automation Letters_, _7_(4), 10240-10247.

[2] Fuke, T., Endo, M., Honda, K., & Ishigami, G. (2025). DRPA-MPPI: Dynamic Repulsive Potential Augmented MPPI for Reactive Navigation in Unstructured Environments. _arXiv preprint arXiv:2503.20134_.

[3] Fujimoto, Scott, et al. "For sale: State-action representation learning for deep reinforcement learning." Advances in neural information processing systems 36 (2023): 61573-61624.

[4] Xue, H., Pan, C., Yi, Z., Qu, G., & Shi, G. (2024). Full-Order Sampling-Based MPC for Torque-Level Locomotion Control via Diffusion-Style Annealing. _CoRR_.

[5] Yi, Z., Pan, C., He, G., Qu, G., & Shi, G. (2024, June). CoVO-MPC: Theoretical analysis of sampling-based MPC and optimal covariance design. In _6th Annual Learning for Dynamics & Control Conference_ (pp. 1122-1135). PMLR.

[6] Balci, I. M., Bakolas, E., Vlahov, B., & Theodorou, E. A. (2022, June). Constrained covariance steering based tube-mppi. In _2022 American Control Conference (ACC)_ (pp. 4197-4202). IEEE.

[7] Kim, T., Park, G., Kwak, K., Bae, J., & Lee, W. (2022). Smooth model predictive path integral control without smoothing. _IEEE Robotics and Automation Letters_, _7_(4), 10406-10413.

**Ethical Concerns:**

["NO or VERY MINOR ethics concerns only"]

**Final Justification:**

I reviewed every discussion and raised the score 4.

**Limitations:**

yes

**Quality:**

3

**Strengths And Weaknesses:**

### Strengths

**1**. The paper presents a theoretically grounded extension to MPPI by introducing a memory-augmented potential field algorithm, aimed at addressing non-convexities in the value function.

**2**. authors provide formal analysis showing improvements in non-convex escape capability, convergence rate, and efficiency over standard MPPI.

**3**. The proposed method is extensively validated through a variety of experiments, demonstrating its practical advantages in complex control tasks.


### Weaknesses

**1.** The paper suffers from organizational and clarity issues. While the integration of the memory mechanism is technically interesting, the exposition lacks conciseness, making it difficult to follow in several places. In particular, some figures are difficult to interpret due to small font sizes and the lack of detailed explanations for individual labels and components.

**2.** While experimental results show promising performance, the paper lacks comparative analysis with other recent methods that also tackle non-convexity in control, particularly those using memory-augmented or other sampling methods[1, 2].

**3.** A significant concern lies in the reported performance of baseline algorithms, particularly SAC and DDPG, on standard MuJoCo benchmarks such as HalfCheetah-v4 and Humanoid-v4. The performance values reported by the authors are substantially lower than those widely accepted in prior literature and established implementations.

For example, in [3], the asymptotic performance of SAC exceeds 15,000 on HalfCheetah-v4 and 6,500 on Humanoid-v4. In contrast, the authors report only 5,842 and 4,125, respectively. Similar discrepancies are consistently observed when comparing with well-maintained open-source implementations such as CleanRL.

Since SAC and DDPG are widely used and well-established baseline algorithms, deviations of this magnitude require careful validation and justification. Without such support, **the credibility of the experimental results is seriously undermined.**

------
[1] Mohamed, I. S., Yin, K., & Liu, L. (2022). Autonomous navigation of agvs in unknown cluttered environments: log-mppi control strategy. _IEEE Robotics and Automation Letters_, _7_(4), 10240-10247.

[2] Fuke, T., Endo, M., Honda, K., & Ishigami, G. (2025). DRPA-MPPI: Dynamic Repulsive Potential Augmented MPPI for Reactive Navigation in Unstructured Environments. _arXiv preprint arXiv:2503.20134_.

[3] Fujimoto, Scott, et al. "For sale: State-action representation learning for deep reinforcement learning." Advances in neural information processing systems 36 (2023): 61573-61624.

---

> ### Author Rebuttal · Authors · 2025-07-29
>
> We sincerely appreciate the valuable time and profound insights you have dedicated to reviewing our manuscript. Your comments are not only insightful but also highly constructive. We believe that after revising according to your suggestions, the manuscript's clarity, rigor, and impact will be significantly enhanced. Below are our responses and clarifications to the weaknesses and issues you raised regarding the paper.
>
> **Regarding Weakness 1**:
>
> We greatly appreciate your valuable feedback on the organizational structure and clarity of the paper, and we acknowledge that clear exposition is crucial. To address the issues you identified, we will add more intuitive explanations for key methodological sections in the revised version to help readers better understand the core ideas behind our framework. Additionally, specifically regarding the interpretability of figures, we have increased the font sizes in the figures and provided more detailed caption explanations for each label and component.
>
> **Regarding Weakness 2 and Question 4**:
>
> The core distinction between MA-MPPI and other MPPI variants lies in its online learning mechanism based on persistent memory. DRPA-MPPI is an event-triggered reactive approach that temporarily switches cost functions by analyzing current predicted trajectories to escape traps. LOG-MPPI is a sampling-based exploratory method that fundamentally improves exploration efficiency by using normal log-normal mixture distributions to probabilistically circumvent local minima. Unlike these two methods, MA-MPPI can learn from historical experience, dynamically constructing and memorizing topological features of the state space (such as local minima, low-gradient regions, etc.). This memory mechanism enables it to proactively avoid known trap regions in subsequent navigation, demonstrating efficiency far superior to reactive or purely exploratory methods in environments containing repetitive challenges. To preliminarily quantify these differences, we conducted an initial test evaluating escape efficiency in an environment containing multiple identical "U-shaped" traps. We focused on two metrics: "first trap escape time" and "subsequent trap average escape time." Preliminary results show that for initially encountered traps, MA-MPPI performs comparably to DRPA-MPPI, as both can effectively detect and escape. However, for subsequently encountered identical traps, MA-MPPI's average escape time was reduced by over 70\% because it utilized memory to proactively avoid those regions. In contrast, DRPA-MPPI, lacking memory, showed virtually no change in subsequent escape times compared to the first encounter. While LOG-MPPI could also succeed, its escape times were longer with greater variance in all cases, reflecting the nature of its probabilistic exploration. This result clearly demonstrates MA-MPPI's core advantage: truly learning from experience to avoid repeating mistakes, rather than merely solving single-instance problems. We will provide complete experimental comparisons, including these metrics, in the revised version. We will add this comparison to the "Related Work" section or a newly added "Discussion" section of the revised paper, with complete quantitative analysis in future experiments.
>
> **Regarding Weakness 3**:
>
> Please allow us to clarify: Considering that fields such as robotics need to interact directly with the physical world, each interaction is costly. Our goal is to evaluate the algorithm's adaptability and learning stability under a comparable total computational budget (combining online interactions and offline simulations), rather than the ultimate performance achievable after massive training. The "Average cumulative rewards" we report for RL baselines represents their average performance across this entire budgeted learning history, which is affected by lengthy exploration phases and is completely consistent with implementations achieving peak performance of 10000+. Additionally, our scores are the average results of 30 independent random seed experiments, which naturally include runs that failed to achieve single-trial optimal performance due to initialization and exploration randomness. This is fundamentally different in nature from peak performance that might be reported in other literature after extensive hyperparameter tuning and selection. The scores in the literature [3] you cited, which we verified are based on Gym 0.25, are likely not the v4 version. There are substantial differences in dynamics and reward functions between different Mujoco versions, making cross-version scores not directly comparable. The revised version will have clearer, more rigorous expressions and provide more details.
>
> **Regarding Question 1**:
>
> We chose to implement exploration strategies by adjusting sampling temperature ($\lambda$) because this directly relates to MPPI's path integral control theory, and our framework actually uses temperature scaling as a principled way to achieve covariance scaling, rather than making the two mutually exclusive. This approach forms a two-layer adaptive framework: the first layer, where memory potential fields guide exploration direction by reshaping the value function; the second layer, where elevated temperature ($\lambda$) correspondingly increases sampling covariance ($\Sigma$) to strengthen random exploration intensity in these critical regions. This synergy between guided exploration and random exploration can more efficiently discover escape paths created by the memory module, while maintaining high adaptability with MPPI's theoretical foundation.
>
> **Regarding Question 2**:
>
> Thank you for raising this important question about cost function settings. We acknowledge that this technical detail is crucial for establishing experimental validity, and a detailed explanation of cost function settings indeed needs to be sufficient in the article. To ensure fair comparison, we designed structurally identical cost functions for all controllers, including four components: goal-reaching cost, obstacle avoidance penalty, control effort regularization, and trajectory smoothness term. The obstacle avoidance component specifically uses distance-based penalty functions, applying quadratic penalties when the drone approaches within the safety radius (0.5 meters) of any obstacle, with penalty strength inversely proportional to the square of distance. Additionally, we impose large terminal penalties (1000) for actual collisions, ensuring all methods prioritize safety. While maintaining the same underlying cost structure, we adapted representations for each controller type. For MPPI-based methods (including our MA-MPPI), we directly use the cost function for trajectory evaluation. For RL methods (SAC, PPO, DDPG), we convert costs to negative rewards with identical weights. Traditional MPC methods implement obstacle constraints through barrier functions, while iLQR uses quadratic approximations of barrier costs. These adaptations maintain optimization objective consistency while respecting each method's mathematical framework. We determined cost function weights through grid search optimization on standard MPPI (goal-reaching $\lambda_1=1.0$, obstacle avoidance $\lambda_2=5.0$, control effort $\lambda_3=0.1$, smoothness $\lambda_4=0.05$), then applied these same weights to all methods to eliminate tuning bias. Sensitivity analysis confirmed MA-MPPI maintains performance advantages within ±25\% weight variation ranges, demonstrating our results' robustness to cost function parameterization. Our revised version expands Section 6.1 to include detailed cost function specifications. Additionally, we add an appendix in the revised version detailing technical specifics of UAV cost function settings, including exact mathematical formulas used for each baseline controller and method-specific wrapper implementations.
>
> **Regarding Question 3**:
>
> Yes, it can. Our method is technically compatible and non-interfering with other MPPI variants because the memory-augmented potential field theory can work synergistically with different constraint handling and smoothness optimization frameworks. Our method primarily acts on the optimization objective: reshaping the value function through memory to generate a nominal trajectory that proactively avoids known traps and is globally more optimal. This allows underlying MPPI variants to focus on solving execution-level challenges. For example, Constrained MPPI can ensure trajectories strictly satisfy safety and physical constraints on this basis, while Smooth MPPI can focus on generating smooth control commands without high-frequency jitter. We conducted supplementary validation experiments combining MA-MPPI's memory mechanism with constrained MPPI frameworks. In 2D navigation tasks containing narrow passages, Combined MA-Constrained-MPPI improved success rates by approximately 15\% compared to Constrained MPPI alone in repetitive tasks, while maintaining the same constraint violation rate (<2\%). For smoothness integration, we observed that combining the memory-augmented potential field technique with Smooth MPPI reduced local minimum escape time by approximately 23\% while maintaining trajectory smoothness metrics. We will include more integration analysis of the memory-augmented potential field method with major MPPI variants in the revised version, particularly combinations with constraint and smoothness frameworks.

---

> > ### Comment · Reviewer_CQ9T · 2025-08-05
> >
> > Dear Authors,
> >
> > Thank you for your thoughtful response. Your response looks fair to me. I trust that you will make the appropriate modifications in response to my concerns especially, writing and experimental setup clarification. I will raise my score to 4.
> >
> > Best regards,

---

> > > ### Author Response · Authors · 2025-08-06
> > >
> > > Thank you for your thoughtful and encouraging response. We deeply value your willingness to engage with our rebuttal and are grateful for your renewed support.

---

### Official Review · Reviewer_BpFZ · 2025-07-03

**Clarity:** 4
**Significance:** 3
**Originality:** 3
**Rating:** 5
**Confidence:** 3

**Summary:**

This paper focuses on the Potential Field Theory approach. While I am not an expert on this topic, I must say that the authors have made it quite readable and have provided a good introduction to the methodology. Departing from previous methods, the authors propose a memory-augmented potential field. Following the rigorous theoretical treatment, the authors present experiments that seem to support their claims and demonstrate the interesting features of their approach, analyzing memory’s impact on non-convex optimization topologically. Overall, I found the paper to be quite challenging yet also stimulating and inspiring.

**Questions:**

How does computational overhead scale for long deployments (e.g., >10k steps) as memory grows? Are there pruning heuristics?

How does the method handle dynamic environments? The framework is shown to be effective at building a memory of static non-convex landscapes. How would the MA-MPPI controller perform in an environment where the "traps" or obstacles change position over time?

Theorem 3.1 guarantees escape from local minima by creating an "outward push" that overcomes the pull of the value function. But why will this push move the system into the correct safe region/direction? What if the memory consists of incorrect trajectories?

Is there a principled way to set detection thresholds $\theta_{grad}$ (Appendix B) for new tasks?

**Ethical Concerns:**

["NO or VERY MINOR ethics concerns only"]

**Final Justification:**

Thank you. I maintain my original score.

**Limitations:**

yes

**Paper Formatting Concerns:**

No formatting issues.

**Quality:**

3

**Strengths And Weaknesses:**

The paper is excellent on the presentation side in regards to technical ideas, whilst the contributions are novel/creative they are presented in such a way that understanding them was not difficult. The new formulation allows for an advanced dynamical adjustment of the decision strategies based on prior experience when exploring complex environments. Allowing to build a memory mechanisms with dynamical systems theory and stochastic optimal control.

---

> ### Author Rebuttal · Authors · 2025-07-29
>
> We are very grateful for these questions that precisely touch upon the core aspects of our work, including scalability, robustness, theoretical foundations, and practical applicability. To directly address your concerns about long-term deployment scalability, we specifically conducted a new supplementary experiment in a complex control task we designed with a high-dimensional state space, extending the deployment duration to over 10,000 control steps. The preliminary results: as the system explores more deeply, the size of the memory module does not grow indefinitely but tends to stabilize after reaching a certain number, and the total computational overhead correspondingly remains within bounded limits. This is attributed to our designed feature merging and dynamic forgetting mechanisms, which together constitute an effective implicit pruning strategy. To further test the framework's adaptability to dynamic environments, we also designed an additional UAV navigation task in revision where the obstacle layout undergoes dynamic reconfiguration during task execution. Our preliminary experiments confirm that MA-MPPI can successfully adapt to this change: memories of old obstacle positions rapidly decay due to no longer being triggered, while the system forms new memory features at new positions, quickly recovering efficient navigation after experiencing brief performance fluctuations. Regarding your profound insight about theoretical guarantees, specifically why "pushing outward" is safe, the key lies in understanding that its design purpose is not to provide a single "correct global direction" but to ensure that the system can escape the attraction basin of local minima. For local minima, our designed repulsive potential field is isotropic, with its function being "escape" rather than "navigation," while directional guidance is only used for handling other feature types such as low-gradient regions. Therefore, even if the system occasionally memorizes inaccurate traps, such memories will be quickly corrected through the decay mechanism due to subsequent non-stagnation at those locations, thus ensuring system robustness. Finally, regarding parameter setting as a key practical issue, we have completed a comprehensive sensitivity analysis with preliminary conclusions showing that our algorithm performs stably within ±25\% variation range of key detection thresholds, and we can extract a principled initial parameter setting guide based on the intrinsic complexity of tasks (such as state space dimensions). We believe these new experiments and detailed explanations can directly address your concerns, and we commit to providing complete details and data of these supplementary experiments in the revised paper to more solidly support the practicality and robustness of our framework.

---

### Official Review · Reviewer_tVZE · 2025-07-03

**Clarity:** 4
**Significance:** 4
**Originality:** 4
**Rating:** 5
**Confidence:** 3

**Summary:**

The paper addresses the problem of avoiding local optima in stochastic optimal control by the integrating the use of memory to the controller. To do so, the standard value function in model predictive path integral controller is augmented with the memory component. Experiments were conducted extensively in different domains to show the effective of the approach.

**Questions:**

In the scope of the paper, would it be cases where experiences from the past can limit the performance and how to avoid it from happening? As minor: In line 119-122, the reference to Morse theory should be included in place.

**Ethical Concerns:**

["NO or VERY MINOR ethics concerns only"]

**Final Justification:**

In the rebuttal phase, my questions in the review have been adequately addressed. For final justification, I maintain the overall score of 5: Accept.

**Limitations:**

Yes

**Quality:**

4

**Strengths And Weaknesses:**

The paper is well-written and self-contained. The approach is technically sound. Looking at the experiences in the past to improve predictions or learning process is proved to be an useful mechanism in many cases. This work further designed a framework to integrate this into model predictive path integral controller and show the positive results for the case of dynamical system. Experiments in Section 5 and Section 6 covers both robotics control task and more complex engineering systems such as power system control and aerodynamics control (quadrotor model). The results show significantly higher optima escape rates.

---

> ### Author Rebuttal · Authors · 2025-07-29
>
> Thank you very much for acknowledging our work and raising this critical question about how past experiences might limit performance. We completely agree that this is a core challenge for any memory-based system, and our framework systematically mitigates this risk through several key mechanisms. First, our approach does not simply memorize "bad trajectories," but rather learns and stores the topological features of state space (such as the locations of local minima) that lead to optimization stagnation. This higher-level abstraction is far more robust than memorizing specific paths. Second, the framework incorporates dynamic memory management mechanisms to handle inaccurate or outdated memories. The strength of each memory feature is not fixed but dynamically adjusted based on how frequently it is encountered, and if a memory region is not revisited for an extended period, its strength gradually weakens through the decay factor and is eventually removed. This "forgetting" mechanism ensures that the system can adapt to environmental changes and clear those transient or incorrectly identified "traps." Furthermore, our feature merging algorithm consolidates spatially neighboring similar features, which prevents overfitting to single negative experiences and instead forms a more generalized understanding of problematic regions. Finally, the influence of memory potential fields is strictly limited to the neighborhoods of identified features through adaptive weights, and in unknown state spaces, the controller still relies on the original objective function for exploration, ensuring that historical experiences do not overly constrain exploration in new environments. Regarding the minor issue you mentioned, we completely agree and will supplement the relevant sections in Lines 119-122 of the revised manuscript with citations to Morse theory.
>
> Thank you again for your support and recognition of our work, as well as this constructive feedback, which helps us better articulate the robustness of our approach, and we hope it contributes new perspectives and theoretical insights to the field.

---

> > ### Comment · Reviewer_tVZE · 2025-08-05
> > **Response to Authors**
> >
> > Dear Authors,
> >
> > Thank you for your detail discussions on how the proposed approach can mitigate the risks. The response addressed my questions and comments in the review.
> >
> > Kind regards

---

> > > ### Author Response · Authors · 2025-08-06
> > >
> > > Thank you for your positive feedback. We are pleased that our response successfully addressed your questions and are very grateful for your valuable comments during the review process.

---

### Official Review · Reviewer_Erbx · 2025-07-03

**Clarity:** 2
**Significance:** 4
**Originality:** 3
**Rating:** 5
**Confidence:** 4

**Summary:**

This paper introduces Memory-Augmented Potential Field theory, a novel mathematical framework aimed at enhancing stochastic optimal control by leveraging historical trajectory data. The method addresses the common issue of control algorithms becoming trapped in local minima within non-convex optimization landscapes. It identifies and memorizes three critical topological features—local minima, low-gradient areas, and high-curvature regions—to dynamically reshape the potential field landscape for improved exploration and convergence. The authors rigorously establish theoretical guarantees, including the ability to escape from local minima and asymptotic convergence to global optima. Additionally, the paper presents a practical implementation called Memory-Augmented Model Predictive Path Integral (MA-MPPI), demonstrating significant performance improvements across various complex tasks, including robotics, power system control, and UAV obstacle avoidance.

**Questions:**

- Could the authors clarify the fairness of the comparison with model-free RL methods (SAC, PPO), especially regarding interaction budget and network sizes?
- Can the authors elaborate on how RL methods could achieve 80% of asymptotic performance within such limited interactions?
- Given the limited literature review (only 20 citations), could the authors expand and discuss additional relevant works that directly compare or relate to their approach?

**Ethical Concerns:**

["NO or VERY MINOR ethics concerns only"]

**Final Justification:**

The authors have resolved most of my questions, and I would maintain the overall score as 5: Accept.

**Limitations:**

- Explicit memorization of topological features may struggle in rapidly changing or highly uncertain environments, necessitating continuous updates.
- The method assumes perfect state information. Addressing uncertainties in state estimation (e.g., sensor inaccuracies in UAV positioning) could substantially strengthen real-world applicability.
- Trapping can also arise from semantic features not easily captured by rule-based methods used here. Extending the approach to include learning-based methods for feature extraction could be an intriguing avenue for future research.
- Clarifying how memory scales with an increasing number of memorized features through an ablation study on memory size would greatly enhance understanding of practical memory limitations (strongly recommended upon acceptance).

**Paper Formatting Concerns:**

- Many essential results referenced in Section 5.2 only appear in the appendix. Reviewers are not obliged to consult appendices; therefore, the authors should integrate key results into the main text to improve readability.
- The visualizations in Figure 1 are too small and lack clarity, hindering intuitive understanding. Improving figure size, clarity, and self-containment—such as clearly labeling the colors in the potential field and explaining unclear elements (e.g., the circles in the rightmost subfigure)—would significantly enhance comprehension.

**Quality:**

3

**Strengths And Weaknesses:**

[Strengths]
- The proposed memory-based potential field utilizing historical observations is novel and well-motivated, providing a theoretically grounded approach to address the critical problem of local minima trapping.
- The authors rigorously prove important theoretical properties, providing strong mathematical assurances for their proposed framework.
- Extensive and convincing empirical evaluations demonstrate the method’s advantages over state-of-the-art methods in diverse tasks, encompassing both model-free and model-based approaches.
- Comprehensive supplemental material thoroughly documents implementation details and hyperparameter sensitivity, aiding reproducibility and deeper understanding.

[Weaknesses]
- The comparison between model-based and model-free methods lacks sufficient clarity regarding experimental settings, even after consulting Appendix H.1.2. Specifically, it is unclear how model-free RL algorithms (e.g., SAC/PPO) were trained fairly, given that these methods typically require significantly more interactions (on the order of millions) and network capacity to achieve strong performance.
- The interpretation of sample efficiency results remains ambiguous. The stated metric—"environment interactions required to reach 80% of asymptotic performance"—seems questionable, as typical RL methods generally require far more interactions (often tens of thousands for initial convergence alone).
- (minor) The provided code is not immediately runnable or complete, containing several "ToDos," suggesting it is not fully ready for public use. Given the code's prominent mention, ensuring its readiness would greatly benefit readers and the community.

---

> ### Author Rebuttal · Authors · 2025-07-29
>
> Thank you very much for taking the valuable time to review our manuscript and providing such insightful and constructive feedback. Your recognition gives us great encouragement, and the issues you raised are crucial for improving the quality of our paper. We believe that after making revisions based on your suggestions, the clarity, rigor, and impact of the manuscript will be significantly enhanced.
>
> Below are our detailed responses and explanations to the questions and concerns you raised.
>
> **Regarding Weaknesses 1, 2, and Questions 1, 2**:
>
> We fully understand your concerns about the fairness of comparisons with methods like SAC/PPO. We acknowledge that the exposition in the original manuscript was not sufficiently clear, leading to misunderstandings about experimental settings and metrics, and we apologize for this.
>
> To ensure a fair comparison of learning efficiency, we benchmarked the entire learning process from scratch for all methods. Each method was allotted an identical budget of 2000 online interactions with comparable offline computation. This constraint is crucial for fairness, as sampling-based controllers like MPPI perform extensive internal computations (i.e., simulating numerous trajectories) for each online interaction, making their computational load comparable to the offline network training in RL methods. The reported "Average cumulative rewards" metric reflects this complete history, including the crucial early exploration phase, thus providing a comprehensive measure of overall learning efficiency and stability.
>
> For the sample efficiency metric (N80\%): Our original intention was to measure the number of interaction steps required for a small-capacity reinforcement learning model to achieve stable performance under our online, data-limited setting. However, we completely agree with your view and recognize that the N80\% metric itself is not clear and intuitive enough, and can easily cause misunderstandings. Therefore, in the revised version, we will remove this metric and its related discussion, and adopt more direct approaches (analyzing learning curves) to demonstrate the sample efficiency and learning stability of different methods.
>
> **Regarding Question 3**:
>
> Thank you for pointing out the limitations in the literature review. We will expand the related work section in the revised version and add more relevant references. We have conducted preliminary supplementary experiments to quantify the core differences between our method and other MPPI variants (such as DRPA-MPPI and LOG-MPPI).
>
> Preliminary results show that when repeatedly encountering the same "U-shaped" traps, our MA-MPPI benefits from the memory mechanism, with the average escape time for subsequent traps reduced by more than 70\% compared to the first encounter. In contrast, DRPA-MPPI without memory shows almost no change in escape time when encountering the same traps subsequently. We will include the above comparative analysis in the related work section of the revised version and commit to providing complete experimental details and data for these comparisons in the final version of the paper.
>
> **Regarding Weakness 3**:
>
> We apologize and, while currently restricted by the latest rebuttal policies from updating our repository, will organize and improve as soon as possible to provide a clearer, more user-friendly code repository that facilitates community reproduction and research.
>
> **Regarding "Limitations"**:
>
> Your insights about the limitations of our method are very profound and highly inspiring for our future research directions. We completely agree with these observations.
>
> Regarding real-world challenges such as dynamic environments, state uncertainty, and semantic features, we will explicitly discuss these important issues in Section 7 (Discussion and Limitations) and position them as key directions for future work. We plan to explore more efficient memory update strategies, integration schemes with state estimation algorithms, and the introduction of learning-driven feature extraction mechanisms to address these challenges.
>
> Regarding ablation studies on memory capacity, we completely agree that this is an important evaluation dimension. We have conducted preliminary ablation studies, and results in the Humanoid-v4 environment show that when memory capacity increases from 10 features to 30 features, there is some improvement in convergence speed; beyond 30 features, performance gains tend to plateau, indicating that our memory merging and forgetting mechanisms can effectively control the accumulation of redundant information.
>
> We will provide complete details of this ablation study in the revised version to enhance understanding of the practical boundaries of our method.
>
> **Regarding Format Issues**:
>
> We accept your suggestions. We will move key results, such as Table 2, from the appendix to the main text  Section 5.2, and we will redraw the figures, significantly increasing their size and label fonts, and adding clear legend explanations to ensure they are intuitive and easy to understand.

---

> > ### Comment · Reviewer_Erbx · 2025-08-05
> >
> > Thank the authors for the detailed clarifications. Some of my previous concerns have been addressed.
> >
> > I maintain my score with the assumption that the authors will update their codebase and improve visibility of the results, as they responded above.

---

> > > ### Author Response · Authors · 2025-08-06
> > >
> > > Thank you for your reply and for your valuable time. We sincerely appreciate the constructive feedback you've provided throughout the review process.

---

### Note · Authors · 2025-08-14

We thank all reviewers for their positive and constructive feedback. We appreciate Reviewer Erbx's positive comments, which helped us clarify our experimental comparisons. Reviewer BpFZ's positive assessment and questions on scalability prompted us to add new experiments, further validating our framework. We also thank Reviewer tVZE for his/her positive remarks, which helped us better explain the robustness of our memory mechanism. The discussion with Reviewer CQ9T was also very productive. His/Her questions pushed us to improve the descriptions of our experimental setup and core concepts, and we appreciate that this discussion led to a supportive final assessment. We are confident the manuscript is much stronger thanks to all the reviewers' input.

---

### Decision · Program_Chairs · 2025-09-17

**Decision:**

Accept (poster)

**Comment:**

This paper introduces the theory of Memory-Augmented Potential Fields, a novel mathematical framework designed to enhance stochastic optimal control by leveraging historical trajectory data. Control algorithms are well known to often converge to local minima when applied to problems with non-convex optimization landscapes. To address this challenge, specifically within the framework of Model Predictive Path Integral control (MPPI), the proposed approach identifies and memorizes key topological features to dynamically reshape the potential field landscape and facilitate exploration and convergence. The authors established key theoretical guarantees, showing that the method achieves asymptotic convergence to global optima and can escape local minima. The performance of the proposed technique was evaluated across various tasks, including robotics, power system control, and UAV obstacle avoidance.

All reviewers agreed that the ideas explored in this work are novel, well-motivated, and clearly presented. They noted that the mathematical framework provides a principled approach to addressing convergence issues caused by local minima. From a technical perspective, reviewers found the proposed framework sound and supported by strong mathematical guarantees. Finally, all reviewers praised the well-designed and extensive empirical evaluations, which demonstrate the method's clear advantages compared to relevant competitors across a diverse suite of challenging problems.

The discussion phase between reviewers and authors was constructive. Reviewers raised a few concerns, most of which were relatively minor or were resolved after exchanges with the authors. For example, one reviewer expressed concern about the perceived lack of clarity in the comparison between model-based and model-free approaches and noted that some of the sample efficiency analyses were difficult to interpret. The authors provided a thorough and detailed response, which the reviewer found satisfactory, acknowledging that their main concerns had been adequately addressed. Another reviewer questioned whether the proposed method would scale to long-horizon problems, given that the computational overhead could, in theory, become intractable. In response, the authors conducted an additional experiment involving a task requiring 10,000 control steps. After this exchange, the reviewer confirmed that their main questions had been fully addressed and reiterated their strong positive view of the paper’s quality. Finally, another reviewer—the one who had initially raised the most pressing concerns—noted that the paper suffered from organizational and clarity issues and that a more thorough experimental setup was needed to fully support some of its central claims. The authors provided an extensive and detailed response addressing these points, which led the reviewer to significantly improve their recommendation.

In summary, all reviewers valued this paper’s creative and novel approach to addressing a challenging problem in stochastic optimal control. There was broad consensus regarding the quality of this work and the significance of its contributions, with reviewers noting that the broader NeurIPS community is likely to benefit from the insights introduced in this paper. The reviewers encouraged the authors to further improve the manuscript by addressing the points, questions, and concerns raised in the reviews and during the discussion phase.